# Getting aligned on representational alignment

Ilia Sucholutsky[1,2*] & Lukas Muttenthaler[3,4,5,6*†]

Adrian Weller[7,21] & Andi Peng[8] & Andreea Bobu[8] & Been Kim[5] & Bradley C. Love[20] & Christopher J. Cueva[8] & Erin Grant[2,9] & Iris Groen[10] & Jascha Achterberg[19‡] & Joshua B. Tenenbaum[8] & Katherine M. Collins[7§] & Katherine L. Hermann[5] & Kerem Oktar[1] & Klaus Greff[5] & Martin N. Hebart[6,22] & Nathan Cloos[8] & Nikolaus Kriegeskorte[11] & Nori Jacoby[12] & Qiuyi (Richard) Zhang[5] & Raja Marjieh[1] & Robert Geirhos[5] & Sherol Chen[13] & Simon Kornblith[14] & Sunayana Rane[1] & Talia Konkle[15] & Thomas P. O'Connell[8] & Thomas Unterthiner[5¶]

Andrew K. Lampinen[5‖] & Klaus-Robert Müller[3,4,5,16,17‖] & Mariya Toneva[18‖] & Thomas L. Griffiths[1‖]

[1]Princeton University [2]NYU Center for Data Science [3]TU Berlin [4]BIFOLD [5]Google DeepMind [6]Max Planck Institute for Human Cognitive and Brain Sciences [7]University of Cambridge [8]MIT [9]UCL [10]University of Amsterdam [11]Columbia University [12]Cornell University [13]Google Research [14]Anthropic [15]Harvard University [16]Korea University [17]Max Planck Institute for Informatics [18]Max Planck Institute for Software Systems [19]University of Oxford [20]Los Alamos National Laboratory [21]The Alan Turing Institute [22]Justus Liebig University Giessen

Reviewed on OpenReview: `https://openreview.net/forum?id=Hiq7lUh4Yn`

## Abstract

Biological and artificial information processing systems form representations of the world that they can use to categorize, reason, plan, navigate, and make decisions. How can we measure the similarity between the representations formed by these diverse systems? Do similarities in representations then translate into similar behavior? If so, then how can a system's representations be modified to better match those of another system? These questions pertaining to the study of *representational alignment* are at the heart of some of the most promising research areas in contemporary cognitive science, neuroscience, and machine learning. In this Perspective, we survey the exciting recent developments in representational alignment research in the fields of cognitive science, neuroscience, and machine learning. Despite their overlapping interests, there is limited knowledge transfer between these fields, so work in one field ends up duplicated in another, and useful innovations are not shared effectively. To improve communication, we propose a unifying framework that can serve as a common language for research on representational alignment, and map several streams of existing work across fields within our framework. We also lay out open problems in representational alignment where progress can benefit all three of these fields. We hope that this paper will catalyze cross-disciplinary collaboration and accelerate progress for all communities studying and developing information processing systems.

---

*Equal contributions as first author. Each block of authors is sorted alphabetically.

†Presently at Aignostics and Helmholtz Munich.

‡Work partly done while an intern at Intel Labs.

§Work partly done while a Student Researcher at Google DeepMind.

¶Presently at Helsing.

‖Equal advising/senior authors.

# Contents

# 1 Introduction

Cognitive science, neuroscience, and machine learning have a long history of studying the kinds of representations that humans, machines, and other biological and artificial information processing systems construct. Numerous factors can affect what representations each system will form, including exposure to and experience with stimuli, diverging training tasks and goals, and differences in architecture – for biological and artificial systems alike. *Representational alignment* refers to the extent to which the internal representations of two or more information processing systems agree. This concept has gone by many names in different contexts, including latent space alignment, concept(ual) alignment, systems alignment, representational similarity, model alignment, and representational alignment (Goldstone and Rogosky, 2002; Kriegeskorte et al., 2008a; Stolk et al., 2016; Peterson et al., 2018; Roads and Love, 2020; Haxby et al., 2020; Aho et al., 2022; Fel et al., 2022; Marjieh et al., 2022; Nanda et al., 2022; Tucker et al., 2022; Muttenthaler et al., 2023a; Bobu et al., 2023; Sucholutsky and Griffiths, 2023; Muttenthaler et al., 2023b; Rane et al., 2023a;b). In addition, representational alignment has implicitly or explicitly been an objective in many subareas of machine learning including knowledge distillation (Hinton et al., 2015; Tian et al., 2019), disentanglement (Montero et al., 2022), and concept-based models (Koh et al., 2020).

While cognitive scientists, neuroscientists, machine learning researchers, and others actively study representational alignment (see Figure 1 for some curated examples), there is often limited knowledge transfer between these communities, which leads to duplicated efforts and slows down progress. We suggest that this, in part, stems from the lack of a shared, standardized language for describing the full spectrum of research on representational alignment. While frameworks such as Representational Similarity Analysis (RSA; Kriegeskorte et al., 2008a) have been broadly adopted as a means of posing comparisons between two systems, they do not capture the full range of work within representational alignment, nor are they applied alike across all disciplines. Ironically, what is needed is greater representational alignment between researchers in the different disciplines that study representational alignment.

In this Perspective, our goal is to provide a theoretical foundation for research on representational alignment across these different disciplines. We conduct a broad literature review across cognitive science, neuroscience, and machine learning (see Section 2), and find that studies of representational alignment generally consist of the same five key components and three objectives. We use this insight to propose a unifying framework (visualized in Figure 2) for describing research on representational alignment in a common language (summarized in Section 4 and Table 2 which illustrates how a broad spectrum of existing studies are easily interpretable when viewed through the lens of our framework). Crucially, our framework provides a way to synthesize insights across disciplines, paving a path towards making progress on the *three central objectives of representational alignment*: measuring alignment, bringing representations into a shared space (which we alternatively refer to as "bridging representational spaces"), and increasing the alignment between systems. Each of these objectives arises in cognitive science, neuroscience, and machine learning (see Figure 1 for an illustrated example of a study from each field for each central objective).

**Objective 1 – Measuring**: The objective of *measuring representational alignment* is typically expressed in terms of determining the degree of similarity between the representational structures of two information processing systems (Shepard and Chipman, 1970; Kriegeskorte et al., 2008a). Thus, measuring representational alignment can offer a principled way to compare two systems at an abstracted, information-processing level, even if those systems appear different at another, often lower, level of detail. This approach can be used to validate one system as a model of another, or to locate cases in which there are differences between two systems. For example, cognitive scientists measure representational alignment between semantic neighborhoods in different languages (Thompson et al., 2020) and different individuals (Marti et al., 2023), as well as between representational maps of musical priors in different cultures (Jacoby and McDermott, 2017; Jacoby et al., 2021b; Anglada-Tort et al., 2023). Neuroscientists measure alignment between humans and non-human primates to establish homology (i.e., the presence of a "common code" in a particular brain region across species) (Kriegeskorte et al., 2008b), measure alignment between a deep neural network model and neural activity recordings to infer which models best capture aspects of perceptual or cognitive processes (Yamins et al., 2014; Khaligh-Razavi and Kriegeskorte, 2014; Yamins and DiCarlo, 2016; Kell et al., 2018; Conwell et al., 2022), and measure alignment between two or more individuals to determine shared motifs in neural activity (Hasson et al., 2004; Stephens et al., 2010; Hasson et al., 2012a) or how synchronization in neural

responses facilitates cooperative behavior (Hasson et al., 2012a; Haxby et al., 2020). Machine learning researchers measure the representational alignment of deep neural networks including computer vision models with humans to test whether these models learn generalizable human-like representations (Langlois et al., 2021a; Sucholutsky and Griffiths, 2023; Muttenthaler et al., 2023a; Ahlert et al., 2024). Typically, the two systems are static, and the data used to measure their alignment is paired (i.e., with the same set of stimuli presented to both systems).

**Objective 2 − Bridging**: The objective of *bringing the representations of two systems into a shared space* (i.e., "bridging" representational spaces) typically involves establishing a correspondence between the representations of the two systems to enable direct comparison. This correspondence unlocks ways of pooling representations across different systems, and of making more directed comparisons than simple measurements of alignment allow.[1] Cognitive scientists aim to compare the representations of different individuals along common dimensions that explain those individuals' behaviors (Wish and Carroll, 1974; Hebart et al., 2020). Neuroscientists align fMRI responses from different individuals into a common space to determine what information is shared across individuals and boost the signal for group-level analyses (Haxby et al., 2011; Chen et al., 2015a; O'Connell and Chun, 2018). Machine learning researchers learn projections from pre-trained image embedding models and pre-trained text embedding models to a joint space in order to enable multimodal prompting (Gupta et al., 2017; Ramesh et al., 2022; Huang et al., 2022). Typically, the two systems are still static, the data may or may not be paired, and the representations from at least one of the systems are projected into a new space.

**Objective 3 − Increasing**: The objective of *increasing representational alignment of two systems* involves trying to make two systems more similar to each other by updating the representations of at least one of the systems. Increasing representational alignment thus can help to make the processing in one system more like another; this can be useful in and of itself (e.g., to improve a computational model of biological system), or as a means to an end (e.g., improved downstream performance). Cognitive scientists try to increase the representational alignment of deep neural networks with humans to better predict human judgments (e.g. Geirhos et al., 2019; Seeliger et al., 2021; Fel et al., 2022; Muttenthaler et al., 2023b). Neuroscientists optimize deep neural networks to predict brain activity to create computational models of brain function (Schrimpf et al., 2018; Toneva and Wehbe, 2019; Schrimpf et al., 2021; Allen et al., 2022; Khosla and Wehbe, 2022; Conwell et al., 2022; Doerig et al., 2023). Machine learning researchers train small, efficient student networks to be as similar as possible to a much larger, more expensive, but highly-performant teacher network (Hinton et al., 2015; Phuong and Lampert, 2019; Tian et al., 2019; Muttenthaler et al., 2024a). Typically, at least one of the systems is dynamic (i.e., it can learn or otherwise update its representations), and the data may or may not be paired data.

Researchers across and beyond these three fields would benefit from progress in each of these areas. We hope that our paper will serve as a call to action for researchers working on representational alignment and catalyze inter-disciplinary collaboration to accelerate progress on these and related problems in the study of information processing systems. To encourage such cross-disciplinary engagement, in addition to proposing a unifying framework for representational alignment in §3 and highlighting key works through the lens of this framework in §4, we also identify key open problems and challenges across disciplines in §5. We believe that resolving these problems would greatly benefit each of the communities that study representational alignment.

## 2   Background and review

Researchers in cognitive science, neuroscience, and machine learning, study various aspects of representational alignment often from differing perspectives, albeit frequently converging on similar techniques. We next review related literature across these fields to motivate our unifying framework (see Table 2) and identify gaps ripe for future work.

---

[1] Though note that some measurements could be seen as *implicitly* bridging into shared representational spaces; e.g., RSA can be seen as bridging from incompatible representation spaces to compatible kernel-like representations which are defined via distances from a set of basis elements.

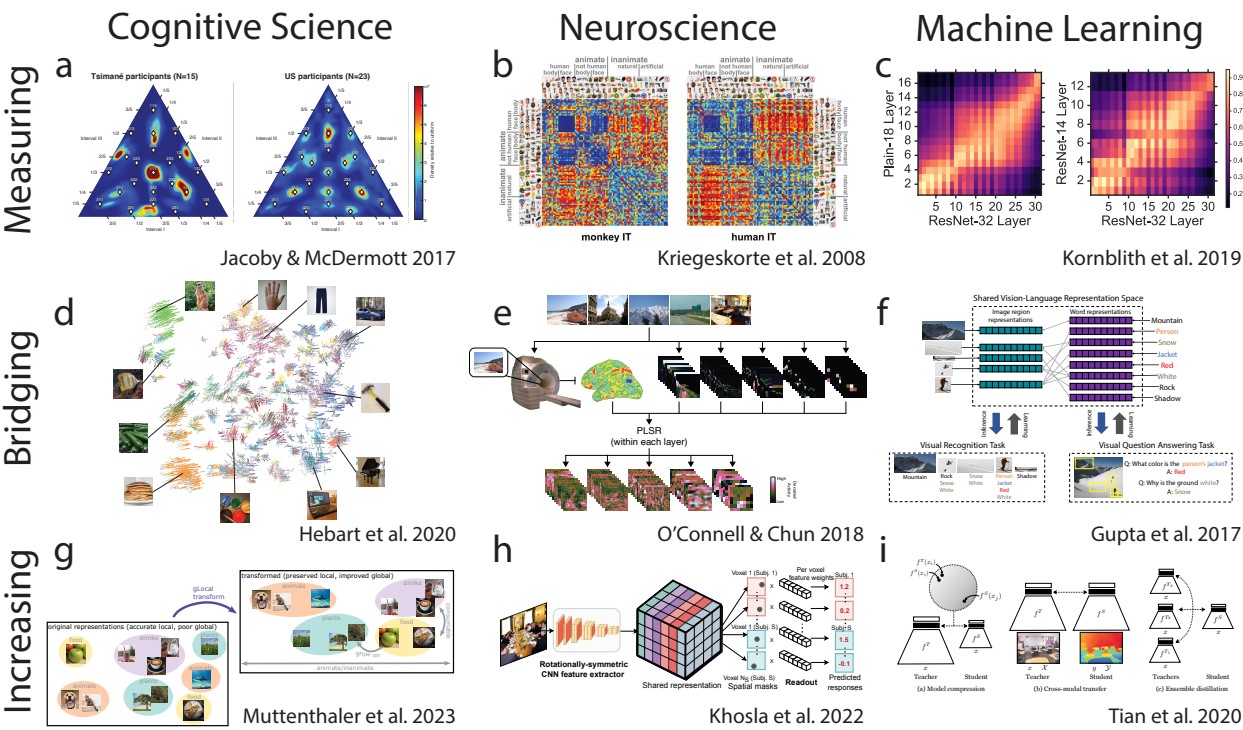

Figure 1: Examples of contemporary representational alignment research in cognitive science, neuroscience, and machine learning. We discuss three types of representational alignment research. *Measuring* representational alignment aims to measure the degree of alignment between two systems as a dependent measure in an experiment (a. measuring cross-cultural similarity in priors for rhythm, Jacoby and McDermott (2017), b. measuring similarity in the representational space of inferior-temporal cortex between humans and monkeys, Kriegeskorte et al. (2008b), c. measuring similarity in the representational space of different neural network architectures, Kornblith et al. (2019)). *Bridging* representational spaces aims to bring representations into a shared space to facilitate some downstream application (d. bridging different individuals' behavior into a common representational space for objects, Hebart et al. (2020), e. bridging fMRI responses and eye movement patterns via alignment between brain responses and neural networks, O'Connell and Chun (2018), f. bridging language and vision via cross-modal alignment of vision and language representations in neural networks, Gupta et al. (2017)). *Increasing* representational alignment aims to update the internal representations or measurements of one system to increase its alignment with another system (g. increasing alignment between human and computer vision model behavior with a semantic grouping task, Muttenthaler et al. (2023b), h. increasing alignment between fMRI responses and neural network activity by direct optimization, Khosla and Wehbe (2022), i. increasing alignment between two neural networks via knowledge distillation, Tian et al. (2019)). (Reproduced with permission from the cited papers.)

## 2.1 Cognitive Science

Whether different people have the same representation of the world is a central question in the cognitive sciences. Questions about potential differences in people's experience of the same stimuli go back to Locke (1847), who considered whether it might be possible to identify whether two people had different perceptual experiences of color. In contemporary cognitive science, questions about whether people share the same representations are prominent in cross-cultural and developmental psychology (Berry, 2002; Miller, 2002; Henrich et al., 2010b). Following the work of Sapir (1968) and Whorf (2012), cross-cultural psychologists ask whether people from different cultures or language groups represent the world in different ways (Berlin and Kay, 1991; Majid et al., 2004; Frank et al., 2008; McDermott et al., 2010; Henrich et al., 2010a; Dolscheid et al., 2013; Majid and Burenhult, 2014; Jacoby et al., 2019; Barrett, 2020; O'Shaughnessy et al., 2023). Likewise, following Piaget (1973), developmental psychologists consider the possibility that children undergo significant conceptual changes as they develop, creating the possibility of incommensurability between the mental representations of children and adults (Carey, 1988). Most of these approaches attempt to *measure*

alignment between different people, or characterize changes in representations over time (e.g., moral concepts; Kohlberg, 1984; Turiel, 2008). Typically, cognitive science approaches these questions by treating humans as black boxes, and indirectly inferring their internal representations and algorithms from their patterns of behavior—methods that have the benefit of transferring well across many systems.

### 2.1.1 Similarity judgments and multidimensional scaling

One tool that has proven useful in exploring these questions is multidimensional scaling (MDS) (Shepard, 1962; 1980). MDS generally uses participants' similarity judgments to embed stimuli into a low-dimensional vector space where the distance between stimuli is inversely proportional to their similarity (Ekman, 1954; Tversky, 1977; Kriegeskorte and Mur, 2012; Peterson et al., 2018; Cichy et al., 2019; King et al., 2019; Hebart et al., 2020), though there exist many popular variants that make additional assumptions – like INDSCAL, which enables the study of individual differences (Wish and Carroll, 1974; Roads and Love, 2024) – and enable exciting applications like mapping the changes in children's representation of numbers as they develop (Miller and Gelman, 1983). Alternatively, methods like second-order isomorphism rely on analyzing the similarity between two sets of relations among different representations of the same objects – e.g., measuring correlation between pairwise similarity judgments for a set of objects and the degree of featural agreement between that same set of objects (Shepard and Chipman, 1970). Similarly, contrast models analyze similarity of relations for systems with discrete properties (Shepard and Arabie, 1979; Tversky, 1977; Tenenbaum, 1995).

Representational similarity methods are powerful because they are compatible with systems that are either continuous or discrete, symmetric or asymmetric, hierarchical or non-hierarchical, etc. (Edelman, 1998) though they do leave open the question of how to assess whether two representations really capture the same information about the world. Goldstone and Rogosky (2002) presented a method for answering this question, based on discovering alignments between two different concept systems that were represented by spatial locations. Crucially, their approach did not require that the matching concepts be identified in advance, rather, they were able to extract plausible alignable concepts at the same time as learning the global mapping between the systems. More recent work demonstrates that natural environments support the alignment of everyday concepts (Roads and Love, 2020) and that children's early concepts appear to exploit these regularities (Aho et al., 2023).

### 2.1.2 Human-machine alignment

Researchers have also begun to use some of these tools to explore the alignment between humans and machine learning systems. For example, Peterson et al. (2018) used similarity judgments to compare representations of images in humans and machines, finding significant correlations between human similarity judgments and the inner product of the activations at the final layer of convolutional neural networks applied to the same images. Curiously, improving model performance (i.e., behavior) does not guarantee an improvement in alignment (Langlois et al., 2021a). In fact, object recognition models that perform better often show worse alignment with human judgments (Roads and Love, 2021) and the representation structure of state-of-the-art language models fails to align with key aspects of human representation structures (Suresh et al., 2023). While Muttenthaler et al. (2023a) found that most computer vision models they tested had low alignment to humans when used out-of-the-box, a learned linear transformation to the human similarity space could minimize that gap — that is, they were able to *increase* representational alignment. However, higher representational alignment does not always translate to improved performance or more aligned behavior. For example, Sucholutsky and Griffiths (2023) discovered a U-shaped relationship between the degree of representational alignment of a teacher and student and their downstream performance on few-shot transfer learning, suggesting that highly aligned and highly misaligned models can generalize effectively from much less data than models with medium degrees of representational alignment with humans. This result, along with evidence that models may unintentionally overfit to the test sets of popular benchmarks and their idiosyncrasies and labeling errors (Recht et al., 2019; Beyer et al., 2020), may explain why performance is not always correlated with representational alignment.

### 2.1.3 Semantic representations

Representational alignment also arises in the study of semantic representations (Rogers and McClelland, 2004; Bhatia et al., 2019). Measuring alignment of the changing representations over learning (and their decay under neurodegenerative disease) between humans and computational and mathematical models (Rogers and McClelland, 2004; Ralph et al., 2017; Saxe et al., 2019) has played an important role in understanding the computational origins of human semantic cognition. Representational alignment has also been used to study the neuro-anatomical basis of these processes (e.g. Ralph et al., 2017), as we discuss below. Recently, research in the alignment of language with other perceptual modalities has been propelled by remarkable advances in large language models which facilitate the quantitative analysis of semantic similarity and provide a rich comparison class against which human behavior can be studied (Bhatia and Richie, 2022; Bhatia, 2023). For example, Marjieh et al. (2023a) showed that embeddings of textual descriptors can be used to construct good proxies for human similarity judgments across different modalities (visual, audio, and audiovisual) and can perform on par with a large set of domain-specific neural networks that directly process the stimuli. This line of work suggests exciting possibilities for bridging between the representational spaces of computational models and humans.

### 2.1.4 Alignment across individual participants' behavior

Research in social psychology and psycholinguistics has also begun investigating alignment across humans. Prior work in these domains, such as the Stereotype Content Model (Fiske, 2018), focused on characterizing group-level phenomena to uncover generalizable insights about how people perceive, understand, and interact with others. For example, distributional semantics investigates bodies of text to understand shared conceptual relations (Boleda, 2020) and average impression ratings are used to study the systemic dehumanization of repressed groups (Haslam and Loughnan, 2014). Recent work has also highlighted individual differences in the structure of representations across people. For instance, differences arise in people's representations of basic semantic categories (Hoffman, 2018), such as animals (Marti et al., 2023), as well as representations of social groups, in the form of stereotypes (Xie et al., 2021), and even complex concepts, such as war and taxes (Brandt, 2022). Differences in representation can have functional consequences for collaboration and communication. For instance, misalignment of word meanings predicts failures of communication across people (Duan and Lupyan, 2023). Strategies for resolving conflict and disagreement hence need to account for both divergence of opinions and alignment of representations (Oktar et al., 2023).

### 2.1.5 Alignment across cultures

More generally, the study of representational alignment across different cultures (Berlin and Kay, 1991; Henrich et al., 2010a; Majid et al., 2004; Majid and Burenhult, 2014; Dolscheid et al., 2013; Barrett, 2020; McDermott et al., 2010; Jacoby et al., 2019; O'Shaughnessy et al., 2023; Frank et al., 2008) plays an important role in cognitive science. Cross-cultural research offers an approach to addressing core problems in cognitive science such as 1) what cognitive and perceptual principles underlie the structure of a given representation (e.g. statistical learning vs. physiological constraints)?, and 2) how is meaning shaped and (mis-)communicated across languages and cultures? As a concrete example of the first problem, Jacoby et al. (2021a) analyzed the representation of musical rhythm in a massive cross-cultural dataset comprising 39 participant groups in 15 countries and showed that participants exhibited a universal inductive bias towards discrete rhythm categories at small integer ratios, though the degree in which specific discrete categories emerged was heavily contingent on culture and the corresponding local musical systems. As for the second problem, Thompson et al. (2020) analyzed the alignment of semantic neighborhoods of 1,010 meanings in 41 languages and showed that semantic domains with high internal structure such as number and kinship tend to be the most aligned, whereas domains such as natural kinds and common actions aligned much less so, suggesting that the meanings of common words are strongly contingent on the culture, geography and history of their users.

## 2.2 Neuroscience

Neuroscientists often measure representational alignment to evaluate accounts of the functional role of neural activity (Turner et al., 2017; Mars et al., 2021). The Representational Similarity Analysis (RSA; Kriegeskorte et al., 2008a) framework developed in cognitive neuroscience was initially motivated by a fundamental challenge

to this mission: How can we compare heterogeneous internal activities across individuals, species, and biological and artificial kinds, especially in advance of a certain account of how these internal activities produce behavior? RSA and similar frameworks applied in neuroscience answer this question by quantifying a particular notion of similarity between neural activity spaces, including heterogeneous ones. The development of these frameworks for measuring alignment between neural activity spaces reflects a longstanding interest in representational alignment within neuroscience; reciprocally, the frameworks themselves have driven substantial new interest in representational alignment within neuroscience (e.g., Dabagia et al., 2023; Schneider et al., 2023). Here, we overview some areas of neuroscience, with a focus on cognitive neuroscience, from the perspective of representational alignment.

### 2.2.1 Alignment across heterogeneous measurements

A foundational problem in neuroscience is defining equivalences across brain regions in different individuals and different species when differing measurement tools are in use. For example, in animals, electrophysiology (e-phys) and microscopy-based methods are commonplace, whereas in humans functional magnetic resonance imaging (fMRI), electroencephalogram (EEG), and other non-invasive methods are common. In RSA, data (neural responses, model activations, behavior, etc.) are converted to a representational dissimilarity matrix (RDM) capturing the pairwise differences between all stimuli in the dataset and abstracting away from the space in which representations are defined. These two RDMs are then correlated to determine whether the two spaces capture the same similarity structurehis technique has been applied broadly in several domains of cognitive neuroscience; one of its earliest empirical applications was to establish structural similarities between rhesus macaque and human inferotemporal (IT) cortex (Kriegeskorte et al., 2008b). The same set of stimuli, consisting of common objects, were shown to monkeys undergoing e-phys recording and humans undergoing fMRI scanning. Using RSA, Kriegeskorte et al. (2008b) found aligned representations between monkey and human IT. RSA-based techniques have also been used for cross-modal alignment of neural responses collected with different modalities. Cichy et al. (2014) derived RDMs over time from human MEG and monkey electrophysiology recordings and RDMs over space from human fMRI responses, then used RSA to align MEG and e-phys signals over time, and MEG and fMRI signals over space and time, capturing spatio-temporal activation patterns not measurable with either modality alone (e.g., Mack et al., 2016).

### 2.2.2 Alignment across individuals

Representational alignment can be used to bridge neural responses across individuals into a common space for subsequent analysis. Standard fMRI preprocessing involves warping to a common anatomical space, but this approach leads to a loss of information due to individual differences in brain morphology. An alternative set of techniques aims to instantiate a joint representational space in order to alleviate the loss of information in anatomical alignment. The most prominent of these functional alignment techniques is hyperalignment (Haxby et al., 2011; 2020), which applies Procrustes transforms to map individual responses to a common space. Variants of hyperalignment further refine the transformation class using functional connectivity or spatial response patterns (Busch et al., 2021). Analogous approaches that make use of contemporary deep learning systems map individual responses to the internal activity space of a deep neural network (Horikawa and Kamitani, 2017; O'Connell and Chun, 2018; Shen et al., 2019; Horikawa and Kamitani, 2022; Sexton and Love, 2022). A distinct approach, shared response modeling, uses a probabilistic framework to isolate individual-specific and shared components in neural responses into a common parameterization (Chen et al., 2015b), which can be applied to improve searchlight analysis of fMRI data (Kumar et al., 2020). Similarly; one can estimate principal components (across subjects) of the responses to common stimuli, then use shared components as a common representational space (e.g. Tuckute et al., 2025). Once all individual responses are aligned in the same representational space, individual responses can be compared or averaged to perform a group-level analysis. Furthermore, secondary models trained on the shared representational space can be applied to brain data in a zero-shot fashion to accomplish decoding feats such as object classification (Horikawa and Kamitani, 2017; Sexton and Love, 2022), eye movement prediction (O'Connell and Chun, 2018), and image reconstruction (Shen et al., 2019).

### 2.2.3   Alignment between brain activity and model systems

Representational alignment between brain regions and computational models has been used to study details of the relationship between computational models and the anatomy and function of neural processes. For example, computational models of human semantic representation (Rogers and McClelland, 2004) have been linked to the neuroanatomy of human multimodal integration — in particular, the idea that anterior temporal regions produce semantic representations that are aligned across modalities (Pobric et al., 2010), which play a key role in binding representations across modalities (Ralph et al., 2017). Neuroscientists have also started to investigate the utility of artificial intelligence (AI) systems as computational models in a variety of cognitive tasks. In vision, early work by Yamins et al. (2014) revealed a hierarchy of alignment between mid- and late-vision regions in rhesus macaques and mid- and late-layers in neural networks optimized for image classification. Contemporaneously, in language research, work by Wehbe et al. (2014b) revealed significant word-by-word alignment between human brain activity evoked by reading a story and representations from early language models (e.g., LSTMs). Progress in AI in the last 10 years has spurred much research in this area, revealing a high degree of brain alignment for more recent models in the domains of vision (Cichy et al., 2016; Zhuang et al., 2021; Konkle and Alvarez, 2022) and language (Khaligh-Razavi and Kriegeskorte, 2014; Schrimpf et al., 2018; Jain and Huth, 2018; Hollenstein et al., 2019; Kubilius et al., 2019; Toneva and Wehbe, 2019; Schrimpf et al., 2021; Toneva, 2021; Caucheteux and King, 2022; Goldstein et al., 2022; Kumar et al., 2023b). Large transformer language models Subsequent work has explored the rich patterns of how alignment often increases with model scale (e.g. Antonello et al., 2023), data diversity (e.g. Conwell et al., 2022), and amount of training (e.g. Pasquiou et al., 2022; Hosseini et al., 2024b).

Recently, work has begun to consider how representational alignment across *models* corresponds with model-brain alignment. For example, Antonello et al. (2021) construct shared representations across models by bridging representational spaces via a common encoder, then explore how this predicts model-brain alignment, and identify a representational component that relates to anatomical organization—suggesting that this shared space has organizational features that may be reflected in brain anatomy. Complementarily, Hosseini et al. (2024a) explores how stimuli where different models are *poorly* aligned tend to be stimuli where none of the models predict the brain well, whereas stimuli where models align tend to produce higher model-brain alignment—suggesting that model-brain alignment is driven primarily by shared "universal" components of representation.

### 2.2.4   Alignment for hypothesis testing

Neuroscientists often use representational alignment to test hypotheses about information processing in the brain. For example, representational alignment has been used to contribute to mechanistic explanations of task-dependent processing in vision (Cukur et al., 2013; Wang et al., 2019) and language (Toneva et al., 2020; Oota et al., 2022) by investigating which of a number of possible candidate hypotheses aligns best with brain responses to a new stimulus. Using these approaches, neuroscientists have hypothesized about the processing of information related to a wide range of stimulus properties, from manipulability and size of individual objects (Sudre et al., 2012) to changes in the content of continuous visual input (Isik et al., 2018). In language, where much current AI progress is due to the development of performant language models, scientists have used representational alignment to claim that a key mechanism that aligns the representations in language models and brains is the next-word prediction objective function (Schrimpf et al., 2021; Caucheteux and King, 2022; Goldstein et al., 2022). However, it is still unclear whether next-word prediction is necessary or simply sufficient to obtain the degree of observed representational alignment (Merlin and Toneva, 2022; Antonello and Huth, 2022), and other scientists have shown that the alignment is due in part to joint syntactic processing (Oota et al., 2023) and lexical-level semantics (Kauf et al., 2023). In the context of these debates, it is often relevant to consider the timecourse of alignment over training—e.g. whether language models can align with human neural representations *before* training (Pasquiou et al., 2022), or after a developmentally plausible amount of language training data (Hosseini et al., 2024b), or whether alignment of vision model representations to brain responses declines after longer training (Scholte et al., 2024).

### 2.2.5  Alignment for stimulus selection or design

Representational alignment can be used to select data for use in stimulus presentations. For example, "controversial" stimuli, which *decrease* measured alignment between models, can be tested on humans or animals to distinguish between competing mechanistic accounts of neural activity (Groen et al., 2018; Golan et al., 2022) or accounts of behavior (Golan et al., 2020). As another example, Tuckute et al. (2023) used representational alignment between human neural representations and transformer language models to design unusual stimuli that drive and suppress activity in the human language network. More recent work has used more sophisticated methods to design stimuli that drive individual voxels based on automatically-derived natural-language hypotheses about their selectivity (Antonello et al., 2024). More generally, there is an extensive literature forming on aligning latent spaces of generative models to reconstruct stimuli from neural data (VanRullen and Reddy, 2019; Mozafari et al., 2020; Ozcelik and VanRullen, 2023; Park et al., 2023; Takagi and Nishimoto, 2023). These works demonstrate that representational alignment can be used for optimization in *stimulus* space. While these methods are only starting to be explored in neuroscience as well as cognitive science, we believe they open exciting new directions for representational alignment research more broadly (see §5.1).

### 2.2.6  Alignment as communication

Spoken language has been construed as a form of representational transmission in which a speaker uses language to instantiate a representation in a listener (Hasson et al., 2012b); this construal is supported by experiments demonstrating representational alignment between speakers and listeners during narration (Stephens et al., 2010; Silbert et al., 2014; Liu et al., 2017). Using fMRI, speaker-listener neural alignment is found across a diverse range of brain regions spanning temporal, parietal, auditory, and prefrontal cortices and only emerges during successful communication, when the listener understands the speaker's utterance (Stephens et al., 2010). This alignment between subjects also appears to be reflected in the contextual representations of modern language models (Zada et al., 2024). In a more naturalistic design, adults' and infants' neural responses were measured simultaneously in an unstructured play environment; in this context, neural alignment, especially in the prefrontal cortex, emerges during joint – but not independent – play (Piazza et al., 2020). Even in the absence of a structured social task, non-verbal social cues such as eye contact and smiling induce neural alignment between two interacting individuals (Koul et al., 2023). Moreover, representational alignment can persist beyond a single interaction; e.g. groups that first saw an ambiguous video independently, then discussed it in a group and arrived at a consensus, produced more aligned neural representations when they watched the video again (Sievers et al., 2024). Perhaps through lasting impacts like these, representational alignment between individuals also appears to potentially play a role in pedagogy, as evidenced by a correlation between improved teacher-student neural alignment and improved learning outcomes (Meshulam et al., 2021; Nguyen et al., 2022; Sucholutsky et al., 2025).

### 2.3  Artificial intelligence and machine learning

Machine learning researchers use representational alignment in diverse ways from measuring the relationship between models to interpreting their performance, bridging between models to fuse (potentially diverse) representation spaces into a single, canonical one, learning more robust and general representations by increasing representational alignment, and mimicking human-like biases and behaviors, among others. In this section, we provide a non-exhaustive overview of some of these use cases.

### 2.3.1  Model-to-model alignment

There has been interest in maximizing model-to-model alignment in the machine learning community for many years (Hinton et al., 2015; Kim and Rush, 2016; Phuong and Lampert, 2019; Cho and Hariharan, 2019; Tung and Mori, 2019). Such questions have taken on a newfound urgency with the rise of large-scale pre-trained foundation models (Caron et al., 2021; Oquab et al., 2024; Roth et al., 2024; Huh et al., 2024; Muttenthaler et al., 2024a), which are difficult and expensive to train but can serve as useful priors for other, smaller models.

In many cases, increasing alignment begins with *measuring* model-to-model alignment — often with RSA — in an attempt to characterize how different learning objectives (Lindsay et al., 2021; Muttenthaler et al., 2023a), tasks (Hermann and Lampinen, 2020), or simply differences in random initialization (Mehrer et al., 2020) may lead to differences among model representations. These differences can potentially be deleterious to reliability, for instance, when one needs to understand when a similar model may fail.

However, differences between models can also be desirable; indeed, there are many cases where one would like to measure and even *decrease* alignment between models. For instance, to use multiple models in an ensemble, one is likely interested in diverse models that have very different representations (Lakshminarayanan et al., 2017; Fort et al., 2019; Pang et al., 2019; Wu et al., 2021). If diversity is not specifically encouraged, different deep learning models end up being highly aligned with each other because they tend to converge to similar local minima (Mania et al., 2019; Geirhos et al., 2020b; Meding et al., 2021; Moschella et al., 2023; Huh et al., 2024).

We remark that there exist a few alternative approaches to alignment for learning joint representation spaces, such as Contrastive Predictive Coding (CPC; Oord et al., 2018) or Joint Embedding Predictive Architectures (I-JEPA; Assran et al., 2023).

**Multimodality**. Combining several input modalities into a single learning system has a long history (Mori et al., 2000). Deep learning allows us to combine neural architectures designed for different input modalities, and to optimize them jointly. For example, an early such model by Karpathy and Fei-Fei (2015) combined a text representation from an LSTM (Hochreiter and Schmidhuber, 1997) with an image representation from a Convolutional Neural Network (LeCun and Bengio, 1998), and jointly optimized them to produce descriptive captions of images. Other models such as CLIP (Radford et al., 2021) explicitly aim to align visual and textual embeddings using a contrastive learning objective (Sohn, 2016; van den Oord et al., 2018). Fusing architectures designed for a single modality can both be used to transform from one modality into another one, e.g., to align visual inputs and their textual descriptions to caption an image (Karpathy and Fei-Fei, 2015; Xu et al., 2015), to learn a combined embedding space for vision and language (Radford et al., 2021; Zhai et al., 2023), to generate images from a textual description (Mansimov et al., 2016; Ramesh et al., 2021; Saharia et al., 2022; Yu et al., 2022) or to combine text, images, and speech into a single prediction model (Kaiser et al., 2017). All of these models go beyond just bridging the representations learned by their constituent sub-modules, but rather fine-tune them to optimize the alignment between them.

The techniques involved in this research are often similar to those we see in related fields. For example, a recent article employed cross-model alignment (Moayeri et al., 2023) to align image representations with text representations. The technique—which essentially boils down to linear regression—is the same as the one often employed in neuroscience when bridging representational spaces, where a linear mapping from one representation space to another is learned from data. Other works in machine learning have used representational similarity itself as *relative* representation space, which can allow translating between the latent spaces of different models with no training (Moschella et al., 2023; Maiorca et al., 2023; Norelli et al., 2022).

**Knowledge distillation**. Knowledge distillation (Hinton et al., 2015; Phuong and Lampert, 2019) is another way of aligning the representation spaces of two models. The goal of knowledge distillation is to distill the (prior) knowledge of a teacher – usually a large model – about a dataset into a student network – usually a smaller model than the teacher. Instead of training the student network on the labels associated with the data, the student is optimized to match the probabilistic outputs (Hinton et al., 2015), the representational geometry (Cho and Hariharan, 2019), or the pairwise similarities (Tung and Mori, 2019) of a (larger) teacher network. Knowledge distillation can be seen as a form of neural compression or a regularization technique. It has seen successes in various fields of ML, such as machine translation (e.g., Kim and Rush, 2016) and Computer Vision (e.g., Park et al., 2019; Cho and Hariharan, 2019). Part of its success is likely attributable to the use of soft labels which have been shown to yield tighter class clusters (Müller et al., 2019) and improved data efficiency (Sucholutsky and Schonlau, 2021; Collins et al., 2022; Sucholutsky et al., 2023; Muttenthaler et al., 2024a) compared to hard labels. In contrast to soft labels, hard labels rigidly assign zero probability mass to all but the correct class. Moreover, the probabilistic outputs of a teacher network convey implicit information about the relationships between the classes in the data rather than serving the purpose

of replacing the zero entries of hard labels with non-zero probabilities that contain no class-relationship information at all (cf., Müller et al., 2019; Muttenthaler et al., 2024b).

### 2.3.2 Learning human-like representational geometries

There has recently been growing interest in the machine learning community in increasing alignment between human and neural network representational spaces (e.g., Peterson et al., 2018; 2019; Attarian et al., 2020; Roads and Love, 2021; Storrs et al., 2021b; Marjieh et al., 2022; Muttenthaler et al., 2023a; Fu et al., 2023) either to obtain a better understanding of the (dis-)similarities between these spaces (e.g., Muttenthaler et al., 2023a; Mahner et al., 2024) or improve the representational structure of neural networks for increasing their generalizability (e.g., Muttenthaler et al., 2023b; 2024a). Muttenthaler et al. (2023b) attempt to increase representational alignment to align the outputs of computer vision models with human odd-one-out choices for the same set of images, thereby altering the original behavior of the models to improve their downstream task performance on various few-shot learning and anomaly detection tasks. Fu et al. (2023) manipulate the representation spaces of neural nets to align their local similarity structure with that of human observers and, as a consequence, improve nearest neighbor retrieval and local structure. Fel et al. (2022) transform the representations of neural networks to better match the visual strategies used by humans, in doing so improving object categorization performance of neural network models.

Although this line of research is still developing, increasing representational alignment offers vast potential in improving the outputs of systems at a relatively low computational cost — learning a linear transformation (Peterson et al., 2019; Attarian et al., 2020; Muttenthaler et al., 2023a;b) or fine-tuning the parameters of an information processing function (Toneva and Wehbe, 2019; Schwartz et al., 2019; Fu et al., 2023; Muttenthaler et al., 2024a; Sundaram et al., 2024) is much cheaper than optimizing these parameters from scratch —- while at the same time contributing to understanding the factors that drive the alignment between systems (Konkle et al., 2022; Fel et al., 2022; Muttenthaler et al., 2023a).

### 2.3.3 Interpretability and explainability

Human-interpretability is often emphasized in efforts to understand neural networks' representation spaces. Much of this work can be understood as attempting to *bridge* between neural network representational spaces and lower-dimensional or conceptually simpler spaces that human researchers can understand. These ideas date back to early representation learning work at the intersection of AI and cognitive science (Hinton et al., 1986), and were reinvigorated by recent findings in representation learning in language and other areas (Baehrens et al., 2010; Bengio et al., 2012; Mikolov et al., 2013a). In particular, the fact that word representation spaces of words learned by predicting co-occurrence (Mikolov et al., 2013a; Pennington et al., 2014) allowed analogical reasoning by simple linear algebra operations (e.g., $king - man + woman = queen$), attracted a great deal of interest and investigations into the statistical or information-theoretic properties that lead to this phenomenon (e.g., see Ethayarajh et al. (2018) for an information-theoretic analysis of vector arithmetic in skip-gram models).

Some efforts have been interested in interpreting the behavior of artificial neural networks at the level of individual neurons (Bau et al., 2017; Olah et al., 2018; Geirhos et al., 2023), while others investigated how to represent and use human-specified concepts in a neural network for post-hoc interpretability (Bach et al., 2015; Samek et al., 2017a; Kim et al., 2018; Lapuschkin et al., 2019; Samek et al., 2019; 2017b). Embedding or learning human-aligned concepts during training has also been an active area of research (Koh et al., 2020; Zarlenga et al., 2022; Fu et al., 2023; Muttenthaler et al., 2024a) as well as discovering new meanings of learned representations using linear vectors (Yeh et al., 2020; Ghandeharioun et al., 2021). Another notable attempt at alignment is mechanistic interpretability – the effort to find a *procedure* in a network (i.e., how a network *does* X rather than just a *concept* Y). For example, finding circuits, using manual hypothesis-driven probing (Olah et al., 2020) or automatically by using techniques like edge attribution probing (Nanda et al., 2023), that qualitatively align with semantic meaning (e.g., curves) could provide valuable insights.

### 2.3.4 Behavioral alignment

*Behavioral alignment* is a form of alignment that aims specifically at aligning the output, or behavior, of one system (often a computational model) with another (often humans). Behavioral alignment can also be seen as an instance of representational alignment, insofar as output behaviors are produced by a representation (e.g., an image embedding) followed by a mapping from representation to output (a softmax layer, a k-nearest-neighbor classifier, etc.) (LeCun et al., 2015). However, the relationship between penultimate representations and behavioral outputs is not one-to-one. Two systems that have very different representations and mappings could still produce the exact same output/behavior (cf. Hermann and Lampinen, 2020), just like very different sorting algorithms (say, "quicksort" and "bubblesort") produce the same output. The reverse is not the case: if there are differences in behavior, this implies differences in either the mapping, the representation, or both. If the mapping is fixed, perfect behavioral alignment is a necessary condition of perfect representational alignment.[2]

Behavioral comparisons between deep neural networks and human perception have seen substantial interest over recent years. For instance, contrasting error patterns of different systems (a behavioral measure), ideally at the fine-grained individual stimulus level (Green, 1964), can be a powerful way to learn about differences in underlying representations (Rajalingham et al., 2018; Geirhos et al., 2020a); and numerous severe differences between neural networks and human perception have been discovered using behavioral experiments (Baker et al., 2018; Peterson et al., 2018; Geirhos et al., 2018; 2019; Peterson et al., 2019; Feather et al., 2019; Jacobs and Bates, 2019; Serre, 2019; Geirhos et al., 2020a; Hermann et al., 2020; Lonnqvist et al., 2020; Funke et al., 2021; Geirhos et al., 2021; Storrs et al., 2021b; Kumar et al., 2021; Abbas and Deny, 2022; Bowers et al., 2022; Dong et al., 2022; Malhotra et al., 2022; Huber et al., 2022; Jaini et al., 2023; Muttenthaler et al., 2023a; Wichmann and Geirhos, 2023; Kumar et al., 2023a; Muttenthaler et al., 2024a). Similarities in behavior can also serve as clues to phenomena happening under the surface both in neural networks and in humans. Rane et al. (2023c) finds a correlation between neural networks' performance in learning visual words and the age at which children acquire those same words, ultimately showing that both are capturing human judgments of how *concrete* or *abstract* a word is. Such behavioral insights often serve as a tool for identifying relevant phenomena that are then further characterized in interpretability and representational alignment work.

Ultimately, different communities weigh output and representational alignment differently. In neuroscience, for instance, representations are often a central research focus, while robotics and reinforcement learning focus more on output. At present, one widely used form of behavioral/output alignment is Reinforcement Learning from Human Feedback (RLHF) (Ziegler et al., 2019; Christiano et al., 2017; Ouyang et al., 2022; Casper et al., 2023), which uses human ratings of an AI system's behavior to learn a separate model which scores new outputs of the system, in an attempt to better align the model's outputs towards those which a human would prefer. However, what the right kind of feedback to elicit from people is for building reward model remains an open question (Casper et al., 2023; Collins et al., 2024a; Wu et al., 2023; Liang et al., 2024).

### 2.3.5 Value alignment

Behavior-focused methods are commonly used for the daunting goal of value alignment (Taylor et al., 2016; Gabriel, 2020; Kirchner et al., 2022): the goal of building a model that aligns with the values of humans, often with the hope that such a model could broadly benefit humanity. Value alignment is notoriously difficult to define and measure. Thus, researchers often evaluate the alignment of model and human behavioral outputs or task performance (Hadfield-Menell et al., 2017; Hubinger et al., 2019), However, monitoring output alignment is insufficient for predicting whether a model will continue to be aligned with humans, or merely appears that way in a constrained evaluation setting, which is important for detecting the emergence of potentially charged behavior (Chan et al., 2023). Similarly, researchers often use behavior-focused methods like RLHF or Constitutional AI (where human oversight is provided via a list of rules or principles) to increase alignment (Christiano et al., 2017; Bai et al., 2022). However, value alignment may be difficult or impossible to achieve through these methods (Eckersley, 2018; Casper et al., 2023).

---

[2]If alignment is not perfect, the relationship between representation and behavior or output depends on the mapping's properties, for instance, whether it preserves monotone relationships. Typically, alignment is best thought of as a spectrum rather than a binary concept.

Could representational alignment offer new possibilities for value alignment? Zou et al. (2023) pursue value alignment via "representation engineering" — finding representational dimensions that are related to valued behaviors like honesty (cf. Burns et al., 2022), and then manipulating those representations to increase the models' tendency to exhibit these behaviors. This strategy hints that aligning the representational structure of models with that of humans could offer benefits for value alignment and all affected downstream tasks—at the very least as pre-conditioning for more targeted interventions.

### 2.3.6 Human-robot interaction

In robotics, we often seek to build robots that perform tasks specified by human users. To do so, robots need to rely on a representation of salient *aspects* of the world that capture the end user's desired task (Bobu et al., 2023). For example, to make a cup of coffee, the robot must learn features that the human user (implicitly or explicitly) cares about, e.g., brand and flavor of coffee as well as the cup orientation and the cup's distance from obstacles, as part of its representation of the task. There are currently two dominant approaches for learning human task representations: one that *explicitly* builds in structures for learning salient task aspects, e.g. feature sets or graphs (Levine et al., 2010; Daruna et al., 2021; Bobu et al., 2021; Peng et al., 2023), and one that *implicitly* extracts them by directly mapping the inputs to the desired robot behavior, e.g. end-to-end approaches like the identity representation (Finn et al., 2016; 2017; Torabi et al., 2018; Xu et al., 2019). Each of these approaches comes with its own set of trade-offs.

On the one hand, specifying explicit task structure is helpful for capturing relevant task aspects like those described above. However, the structure baked in explicitly is *useful only if correct*: without the right inductive bias, robots may misinterpret the humans' guidance for the task or execute undesired behaviors (Bobu et al., 2020). On the other hand, neural networks can implicitly learn task structure in a manner that is faster and less burdensome on the designer, albeit while potentially containing irrelevant information in their representations and correspondingly capturing spurious correlations (Zhang et al., 2018; Rahmatizadeh et al., 2018; Rajeswaran et al., 2018). Recent trends to address this tendency include *feature subset selection methods* (Cakmak and Thomaz, 2012; Bullard et al., 2018; Luu-Duc and Miura, 2019), clever *ways to efficiently collect human data* (e.g., via YouTube or VR) or *reuse past data sets* from the robot's lifespan (Baker et al., 2022). However, there is still no guarantee that these data will be representative of the end user's behavior. Rather than treating humans as static data sources, these methods may benefit from including them as (weak) supervision signals in the alignment process.

## 3 Framework for representational alignment

As we have illustrated, representational alignment is an active and fruitful area of research. However, analyzing the literature from each of the three fields reviewed above makes it clear that representational alignment is a fragmented area of research, reminiscent of the Tower of Babel. Disparate definitions and insufficient knowledge sharing across fields have led to the rediscovery of the same ideas under different names, the repetition of similar mistakes, and underutilized opportunities for cross-disciplinary collaboration. To unify these fragmented communities, we propose a general formalism of representational alignment – a lingua franca that we hope will accelerate progress on open problems in representational alignment research.

### 3.1 High-level overview

Conceptually, we propose that there are five major components to most studies of representational alignment that researchers have control over (see Figure 2 for a schematic description):

(a) The *data* used for alignment, which could be sensory data (a subset of a stimulus space, such as an image set for vision) or higher-level cognitive content. Throughout this paper, we assume that the data is a static sample (e.g., simple stimuli like images). However, this framework can be generalized to cases where systems interact dynamically with an environment, in which case data is replaced by environment states.

(b) The *systems* whose representational alignment is being measured (e.g., humans, animals, deep neural networks, etc.). A system interfaces with the data. This interface is partially controlled by the experimenter (i.e., the experimenter can choose which stimuli to present and how to present them)

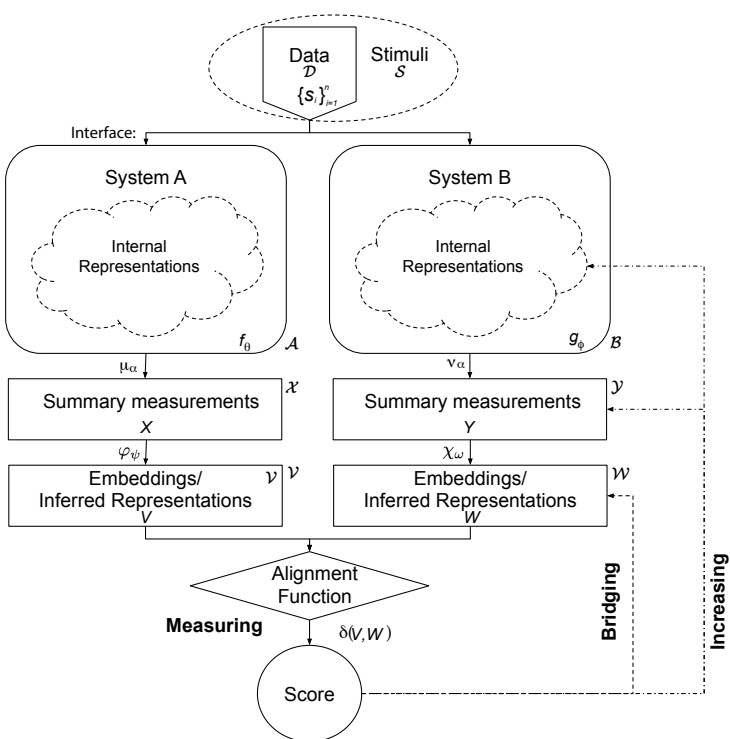

Figure 2: A general framework for conducting and describing representational alignment research. Most studies of representational alignment involve five components that researchers can control: **data** is presented via interfaces to the two **systems**. The systems form internal representations of the data and researchers take **measurements** of the systems and map them to some **embedding** space to try to infer the representations. An **alignment function** is then applied to those inferred representations to compute a single alignment score. These studies typically have one of three objectives: **measuring** the representational alignment between the two systems (i.e., the alignment score), **bridging** between two different representational spaces by finding a shared embedding space, or **increasing** the representational alignment between the two systems either by updating their internal representations (e.g., via learning) or how they are measured.

and partially a component of the system (e.g., for humans this can be the periphery of the visual system). Once the stimulus is internalized in the system (e.g., as neural activity in the human brain), it forms an internal representation (for example, the neuronal activity through the entire brain during preconscious processing). The internal representations of many system states are latent. For human participants, this might be the latent state of their brain as they view an image. In the case of a machine learning system, however, the system can be accessed in principle. An example would be the entire network activation pattern in response to a given stimulus. While we assume that all systems of interest in representational alignment studies can take data as input and form representations of it, we note that in some cases those systems may also have intrinsic or extrinsic objectives that require them to produce outputs (e.g., when the study involves monitoring a system while it performs a task like classification), that those outputs may in some cases affect the data distribution (e.g., by acting on the environment as mentioned above), and that the objectives themselves may affect the internal representations (e.g., task-dependent representations).

(c) The *measurements* that are being collected about each of the systems (e.g. behavioral similarity judgments, activation of a region for fMRI, hidden layer activations for a neural network, etc.). Note that the process of measurement also includes the potentially-different processes required for presenting stimuli to the two systems (e.g., playing audio to a human, versus presenting its spectrogram to a convolutional network).

(d) The *embeddings or inferred representations* that are being extracted or (re)constructed from each system.

(e) The *alignment function* that is being used to measure the degree of alignment between the embeddings.

Studies focused on measuring alignment typically just involve computing an alignment score from the alignment function. Meanwhile, studies focused on bridging representational spaces or increasing alignment usually involve using this score as a feedback signal on how to update the embedding function (in the bridging case) and the internal representations or their measurements (in the increasing case). We visualize this framework in Figure 2.

As a concrete example, consider the work by Kriegeskorte et al. (2008b) highlighted in Panel b of Figure 1. Say we want to measure the representational alignment of two *systems*: a rhesus macaque monkey and a human. In this case, the *data* over which we want to measure alignment might be a collection of scene images. In both monkeys and humans, the state of the two systems would be the activation pattern in all the neurons while they observe the image. This state cannot be directly accessed, but only through *measurement*. For the monkey, this could be the neural responses in the inferotemporal cortex measured with electrophysiology, and for the human, we could define it as neural responses in the inferotemporal cortex measured with fMRI. A widely used summary statistic of the joint representation of all stimuli is the representational dissimilarity matrix (RDM), which defines the representational geometry for each system. The RDM contains the pairwise distances between the activity patterns representing the stimuli, and provides an embedding in which representational geometries can be compared. The RDM comparator (or *alignment function*) can be the cosine similarity, a correlation coefficient, or a metric such as the angle two RDMs span. This approach is known as representational similarity analysis; e.g., Kriegeskorte et al., 2008a; Diedrichsen et al., 2020; Schütt et al., 2023).

We believe that our framework provides a simple, general language for clearly communicating the methodology and results of representational alignment studies in a way that is accessible to many researchers. In Table 2, we present diverse examples of literature from various fields summarized by the components of the framework. The remainder of this section goes into more detail on how to mathematically formalize descriptions of each of the components and decisions that go into a study of representational alignment. We encourage researchers to use our framework when formally describing their representational alignment studies to help others understand the exact details and support reproducibility. In Section 4, we lay out in detail how the nine highlighted examples from Figure 1 can be described in the language of our formalism.

### 3.2 Formalizing representation spaces

Figure 2 shows a schematic description of our framework which contains the following components:

**Data**. Let $\mathcal{D} \coloneqq \{s_i\}_{i=1}^n$ be a dataset of $n$ trials, where each $s_i \in \mathcal{D}$ is a stimulus that can be processed by any information processing function. Note that a dataset is not restricted to a set of single elements. Each element by itself can be either an image, a set of images (e.g., triplets), a string, a sequence (of strings or other realizations of time steps), a video (or frame thereof), etc. In practice, we note that systems can interact with the environment (for example, in the case of an agent in a game environment) in which case the data is the states of the environment and is dynamic rather than static. Most of the case studies in this paper concern the simplified case in which the systems do not modify the environment.

**Systems**. We assume that there exist two systems $A$ and $B$, which can be described in terms of functions that map inputs ($s_i$) to their internal states $f_\theta \coloneqq S \mapsto \mathcal{A}$ and $g_\phi \coloneqq S \mapsto \mathcal{B}$, where $\mathcal{A}$ and $\mathcal{B}$ denote the space of all possible states of systems $A$ and $B$, respectively. For notational simplicity, we abstract away the interface layer, which may affect how the stimuli are presented to each system, as part of $\theta$ and $\phi$. We also note that in some studies, the systems may not only be (passively) processing the stimuli but will be actively engaging with them in a certain task (i.e., producing outputs). Upon performing this task, the systems may modify the environment from which the data is drawn. For simplicity, we treat the environment as stationary and assume that the task context is part of the stimuli.

**Measurements**. For each of the systems $A$ and $B$ we obtain a summary of the measurement $X \coloneqq \mu_\alpha\left(f_\theta\left(s_i\right), \ldots, f_\theta\left(s_n\right)\right) \in \mathcal{X}$ and $Y \coloneqq \nu_\beta\left(g_\phi\left(s_i\right), \ldots, g_\phi\left(s_n\right)\right) \in \mathcal{Y}$, respectively. This is obtained by

sequentially applying the functions $f_\theta$ and $g_\phi$ (returning the state of the systems for each of the stimuli) to all of the $n$ trials and then passing the output through some (possibly) parameterized functions $\mu_\alpha$ and $\nu_\beta$. The parameters $\alpha$ and $\beta$ will often reflect hyperparameters of the measurement process (e.g., in machine learning this parameter could specify which layer activations are being measured from; in human fMRI this can represent parameters of the scanning procedure as well as parameters of processing the raw fMRI data). However, in some cases (typically in machine learning), we simply directly use the entire internal state, and thus $\mu_\alpha = \nu_\beta = \mathbb{1}$ are the identity maps. In this case, $X \coloneqq (f_\theta(s_i), \ldots, f_\theta(s_n)) \in \mathcal{X}^{n \times p}$ and $Y \coloneqq (g_\phi(s_i), \ldots, g_\phi(s_n)) \in \mathcal{Y}^{n \times d}$ are the two-dimensional arrays of stacked measurements of lengths $p$ and $d$ respectively.

**Embeddings**. To map categorical behavior to a continuous number space, denoise a set of high-dimensional measurements (that potentially have a low "signal-to-noise" ratio), or essentially any other reason for why we would need a mapping from the output space (e.g., neural activity) of the information processing functions to another — possibly lower-dimensional — embedding space (e.g., real-numbered values), we can optionally define a function that transforms the measurements into an embedding space where similarity can be quantified. We assume the existence of two embedding functions, $\varphi_\psi \coloneqq \mathcal{X} \mapsto \mathcal{V}$ and $\chi_\omega \coloneqq \mathcal{Y} \mapsto \mathcal{W}$, which can be either linear or non-linear. We also assume that these functions have two optionally learnable arrays of parameters $\psi$ and $\omega$. [3] We emphasize that the embedding function(s) are not necessary but may be advantageous in specific situations. One such scenario includes *increasing representational alignment* (cf., Muttenthaler et al., 2023a;b, see §4 for further examples where this may be desirable). Note that if we do not have an embedding step we can simply assume $\varphi_\psi = \chi_\omega = \mathbb{1}$ are the identity map and do not change the summary measurements.

For simplicity, we consider flattening the representations in all stages into vectors (denoted as lowercase letters in boldface). However, we emphasize that in general, the measurements can have any shape and type — e.g., they may be matrices, graphs, programs, or strings — as long as the two sets of measurements admit an appropriate measure of alignment.

### 3.3 Measuring alignment

There exists a function $\delta : \mathcal{V} \times \mathcal{W} \mapsto \mathbb{R}$ that we can apply to the embedded vectors $\boldsymbol{v}$ and $\boldsymbol{w}$ such that $\delta(\boldsymbol{v}, \boldsymbol{w}) \in \mathbb{R}$ yields a scalar value that quantifies the degree of alignment. For simplicity, we define $\delta(\boldsymbol{v}, \boldsymbol{w}) = \Delta_{\boldsymbol{v}, \boldsymbol{w}}$ to be a dissimilarity measure where $\Delta_{\boldsymbol{v}, \boldsymbol{w}} = 0$ implies that $\boldsymbol{v} = \boldsymbol{w}$, and, therefore the embedding vector $\boldsymbol{v}$ is fully aligned with $\boldsymbol{w}$.

**General conditions**. The following conditions have to be satisfied for any function $\delta$ that measures representational alignment.

- *Measurable.* $\delta$ must be a measurable (dis-)similarity function. However, we do not restrict $\delta$ to be a metric because symmetry is not a necessary condition to assess the alignment between two embedding spaces.

- *Scalar-valued.* To meaningfully quantify representational alignment, we restrict $\delta$ to map to a scalar. Hence, $\delta : \mathcal{V} \times \mathcal{W} \mapsto \mathbb{R}$. For simplicity, in the remainder of this section, for $\boldsymbol{v}$ and $\boldsymbol{w}$, we focus on (flattened) vector representations.

- *(Dis-)similarity-quantifying.* The scalar-valued output of $\delta$ is required to quantify a (dis-)similarity. For convenience, we generally use the notation of a dissimilarity measure, where $\delta$ has a lower bound at zero at which the two embedding spaces are equivalent. Hence, $\delta : \mathcal{V} \times \mathcal{W} \mapsto [0, \infty) \subset \mathbb{R}$. The advantage of a dissimilarity measure is that it can be viewed as an error function or a loss that can be minimized. However, alignment functions could also measure similarity (see §3.3.1).

In the following, we will elaborate on properties of alignment functions we think are useful to distinguish from one another. We distinguish *similarity-quantifying* from *dissimilarity-quantifying*, *descriptive* from *differentiable*, and *symmetric* from *directional* alignment. A valid alignment function must satisfy at least

---

[3]Dimensionality reduction techniques such as SVD or PCA can serve as valid (optional) embedding functions even though they do not consist of any learnable variables. However, one may be interested in learning a particular (non-)linear transformation for which learnable variables are necessary (e.g., to increase alignment).

one of two properties that we contrast in each case. It must be (dis-)similarity quantifying; descriptive, differentiable, or both; and symmetric or directional. We list examples of alignment functions in Table 1 but a more in-depth survey can be found in (Klabunde et al., 2023).

### 3.3.1 Similarity or dissimilarity quantifying

Any alignment function $\delta$ has to quantify the (dis-)similarity between two representations of a set of stimuli (or pieces of cognitive content). Although any similarity can in principle be transformed into a dissimilarity and vice versa, *similarity-quantifying* and *dissimilarity-quantifying* alignment functions have distinct advantages and disadvantages.

**Similarity-quantifying**. Similarity-quantifying alignment functions are often used for describing the relationship between two sets of measurements $\mathcal{X}$ and $\mathcal{Y}$. Among the set of similarity-quantifying alignment functions exist functions that are bounded in both directions. The upper and lower bounds provide reference points that can ease interpretation. Examples include the Pearson correlation, the Spearman rank correlation, the cosine similarity, and any centered or normalized inner product. For these function, we have $\delta : \mathcal{V} \times \mathcal{W} \mapsto [-1, 1] \subset \mathbb{R}$. The bounded nature of these functions renders them particularly insightful for describing a relationship between representations, as its output is easily interpretable.

**Dissimilarity-quantifying**. For all dissimilarity-quantifying alignment functions, $\delta : \mathcal{V} \times \mathcal{W} \mapsto [0, \infty) \subset \mathbb{R}$, holds. That is, dissimilarity-quantifying alignment functions have a lower bound at 0, where we know that two representation spaces are equivalent. However, it is difficult to put an upper bound on these functions. Thus, dissimilarity-quantifying functions can be more difficult to interpret. Information-theoretic measures such as the cross-entropy or relative entropy and $\ell_p$-norms of the difference between two embedding vectors $\boldsymbol{v}$ and $\boldsymbol{w}$, e.g., $||\boldsymbol{v} - \boldsymbol{w}||_2^2$, are common examples of dissimilarity-quantifying alignment functions (e.g., McClure and Kriegeskorte, 2016). Although their outputs can be difficult to interpret and are not recommended to (merely) describe the relationship between two sets of measurements, they are useful error functions that can be minimized by gradient descent. In addition, it is possible to use a dissimilarity-quantifying function (e.g, cross-entropy) to maximize a similarity-quantifying function (e.g., cosine similarity) as is often done in contrastive representation learning (Chen et al., 2020; Radford et al., 2021; Muttenthaler et al., 2023b). Similarity quantifying functions that have been transformed into distances, such as the cosine distance—or, equivalently, one minus the Pearson correlation coefficient $(1 - \rho)$—are better suited to *measure* representational alignment. These distances are bounded in both directions with a minimum at 0 and a maximum at 2. This makes them easier to interpret than information-theoretic measures or $\ell_2$-norms, which have no clear upper bound. However, they are not as convenient for *increasing* the degree of representational alignment between information processing systems because it is difficult to use them directly for optimization.

### 3.3.2 Descriptive or differentiable

An alignment function must be *descriptive* or *differentiable* or both. These properties are not mutually exclusive, but in general, we either want to use $\delta$ for describing or increasing representational alignment.

**Descriptive**. A *descriptive* alignment function does not need to be differentiable. Such a function mainly serves to quantify the (dis-)similarity between the two sets of measurements $X$ and $Y$. Hence, descriptive alignment functions are used when researchers aim to *measure* alignment and establish the conditions and system setups that cause representational alignment to emerge rather than aiming to *increase* alignment (see §5 for a more detailed discussion). Descriptive alignment functions are often *symmetric*, as it is desirable to obtain the same measurement of representational alignment if we change the order of the representations: $\delta(\boldsymbol{v}, \boldsymbol{w}) = \delta(\boldsymbol{w}, \boldsymbol{v})$. An example of a descriptive alignment function used in Representational Similarity Analysis (Kriegeskorte et al., 2008a) is the rank-correlation between RDMs as measured by Kendall's $\tau_a$ (Nili et al., 2014) or $\rho_a$ (Schütt et al., 2023)). Rank correlation is attractive for model-comparison in computational neuroscience because it is invariant to nonlinear monotonic transforms of the RDMs, but it is not differentiable. A descriptive and differentiable alternative would be the Pearson RDM correlation coefficient.

**Differentiable**. The objective to *increase* alignment of a model representation to another model or a brain region motivates the use of a differentiable alignment function. Generally, any differentiable alignment function can be regarded as an error function or loss that can be minimized, such that $\mathcal{L}_{\text{alignment}} \coloneqq \delta(\boldsymbol{v}, \boldsymbol{w})$. If

we want to minimize $\mathcal{L}_{\text{alignment}}$ using a gradient, then $\delta$ must be restricted to the set of differentiable functions over the embedding spaces $\mathcal{V}$ and $\mathcal{W}$. For all differentiable alignment functions, we consider the settings of *representational transformation* and *representational fine-tuning*, respectively, to minimize $\mathcal{L}_{\text{alignment}}$.

*Representational transformation*: Representational transformation refers to the case where a model's parameters are frozen and a transformation of its representation is learned as an add-on to the model. An example is the use of linear encoding models fitted to map from neural network model representations to single-neuron responses measured in animals in neuroscience. Representational transformation requires choosing a level of flexibility for the transformation. Although taking the representation spaces as is (without any transform) may be descriptive (especially in the field of Machine Learning (c.f., Muttenthaler et al., 2023a)), manipulating them allows us to compare spaces that are less obviously similar (e.g., by ranking which ones are *relatively* more similar to each other). Thus, there exists a spectrum of transformations, ranging from the identity function (i.e., no transformation), over linear transformations, up until non-linear functions under a constraint such as Lipschitz continuity, weight bounds, or anything else that constrains the output space of the transform to not move too far from the original space.

In representational transformation, we consider two sets of stacked embedding vectors $\boldsymbol{V} \coloneqq (\boldsymbol{v}_1, \ldots, \boldsymbol{v}_n)^\top$ and $\boldsymbol{W} \coloneqq (\boldsymbol{w}_1, \ldots, \boldsymbol{w}_n)^\top$ to be fixed and immutable tensor representations for the $n$ measurements in the data. Here, we do not need access to any of the two sets of source parameters $\theta$ or $\phi$. We learn a transformation $h_\Omega(\varphi_\psi(\boldsymbol{x}_i))$ for one of the two embedding spaces. Here, for simplicity, we choose the representation space $\boldsymbol{V}$. Hence, we are interested in all first-order derivatives, $\nabla \mathcal{L}(\Omega)$, where we optimize the (bounded) parameters $\Omega$ of the transformation by solving the following minimization problem,

$$\arg\min_{\Omega} \mathcal{L}_{\text{alignment}}\left(h_\Omega(\boldsymbol{V}), \boldsymbol{W}\right)$$

In this case, the alignment function is defined to be a dissimilarity measure that can be minimized and used as an error function rather than a similarity measure that has to be maximized.

*Representational fine-tuning*: In representational fine-tuning, we are interested in differentiating through the entire model and update its parameters. Examples include the student-teacher setup in machine learning (e.g., Hinton et al., 2015; Tung and Mori, 2019; Oquab et al., 2024; Muttenthaler et al., 2024a) and nonlinear systems identification approaches in computational neuroscience (e.g., Wehbe et al., 2014a; Fyshe et al., 2015; Seeliger et al., 2018; Toneva and Wehbe, 2019; Schwartz et al., 2019). To perform representational fine-tuning, two conditions have to be satisfied:

1. *Source parameter fixing*: First, we have to fix one of the two sets of parameters $\theta$ or $\phi$, which can be seen as a special case of directional alignment (see below). Here, only one of the sets of measurements $X$ or $Y$ is subject to change, and the other set remains unaltered.

2. *Source parameter availability*: Second, the parameters of the *sources* $\theta$ or $\phi$, depending on which of the two sets we want to fix, have to be readily available. That is, we need access to the set of parameters that we want to update. Although theoretically possible $\forall f \in \mathcal{F}$, in practice, it is unlikely to have access to the synapses of a human or monkey brain after obtaining measurements from them. Thus, this step is relevant only when the goal is to alter the parameters of an artificial intelligence system.

Let us assume that both of the above conditions are met. We fix the parameter set $\phi$, assume access to $\theta$, and evaluate the dissimilarity of $V$ from $W$. That is, we want to differentiate through $\delta$, $\varphi_\psi$, and $f_\theta$ to minimize $\Delta_{V,W}$ and consequently updating the source parameters $\theta$. As such, we are interested in the first-order derivatives with respect to all of those learnable variables. Note that without a restriction in the mapping function representational fine-tuning is not particularly useful if we are interested in whether two sets of measurements $X$ and $Y$ are (dis-)similar because, in high-dimensional spaces, it is likely that there exists a non-linear transformation (e.g., a multi-layered neural network) that can map one space to the other. For representational fine-tuning to be useful, we must test the *generalizability* of the learned mapping to held-out measurements of the target system (here, $Y$), thereby satisfying at least one of the following two conditions

(a) *Few-shot fine-tuning.* We must limit the number of training examples used for fine-tuning the set of parameters. So, if $n$ denotes the number of training examples used for fine-tuning, $n$ should be small; how small exactly depends on the particular task and research question.

(b) *Regularization.* We must put an upper bound on the quantity $||\theta - \theta^*||_2$ such that $\sup ||\theta - \theta^*||_2 < \epsilon$, where $\theta$ is the set of original source parameters and $\theta^*$ is the set of fine-tuned source parameters and $\epsilon$ is a small real-numbered value. That is, we do not want the fine-tuned parameters to move too far away from the original source parameters.

Differentiable alignment functions are specifically of interest for the goal of *increasing* alignment but under certain conditions may also be useful for *bridging* the representation spaces of systems.

### 3.3.3 Symmetric or directional

An alignment function $\delta$ can either be *symmetric* or *directional*; a function cannot be both at the same time. We recommend symmetric alignment functions over directional alignment functions for describing the relationship between two information processing systems if the goal is "just" to *measure* representational alignment rather than bridging their representation spaces or increasing their alignment.

**Symmetric**. For any *symmetric alignment* function, $\delta(V, W) = \delta(W, V)$ must hold. Changing the order of $W$ and $V$ as inputs to $\delta$ is not allowed to change the (dis-)similarity between the embedding spaces $W$ and $V$. Symmetric similarity functions may be desirable for describing the relationship between $\mathcal{X}$ and $\mathcal{Y}$ rather than optimizing for aligning the two spaces. Examples of symmetric alignment functions that are widely used are the inner product, the cosine similarity, or the Pearson correlation, of which the latter two are modified versions of the former.

**Directional**. *Directional alignment* functions define *alignment in terms of one space*. For these functions, $\delta(V, W)$ has to be defined in terms of one of the two embedding spaces $V$ or $W$. Hence, any directional alignment function either measures the dissimilarity of $V$ from $W$ or, the other way around, it measures the dissimilarity of $W$ from $V$. Most information-theoretic measures are directional alignment functions of which common examples are the discrete versions of the cross-entropy and the relative entropy (or KL divergence), where discrete KL divergence is defined as

$$\delta\left(\sigma(\varphi_\psi\left(\boldsymbol{x}_i\right)), \sigma(\chi_\omega\left(\boldsymbol{y}_i\right))\right) := \mathrm{KL}\left(\sigma(\varphi_\psi\left(\boldsymbol{x}_i\right)), \sigma(\chi_\omega\left(\boldsymbol{y}_i\right))\right) := -\sum_{j=1}^m \sigma(\varphi_\psi\left(\boldsymbol{x}_i\right))_j \log\left(\frac{\sigma\left(\chi_\omega\left(\boldsymbol{y}_i\right)\right)_j}{\sigma\left(\varphi_\psi\left(\boldsymbol{x}_i\right)\right)_j}\right),$$

where $\sigma : \mathcal{V} \cup \mathcal{W} \mapsto \mathbb{R}^k$ with $\left\{\sigma\left(x\right) \in \mathbb{R}^k : x_0 + \cdots + x_{k-1} = 1, x_i \geq 0 \text{ for } i = 0, \ldots, k-1\right\}$ is a function that transforms the embedding representations into discrete probability distributions (e.g., softmax). Here, $\sigma(\varphi_\psi\left(\boldsymbol{x}_i\right))$ and $\sigma(\chi_\omega\left(\boldsymbol{y}_i\right))$ must have the same shape. Due to their unbounded nature, information-theoretic directional alignment functions are generally not recommended for describing the relationship between $V$ and $W$ because they are difficult to interpret (see §3.3.1). However, they are useful error functions for minimizing the dissimilarity between two sets of representations and therefore often used for solving general machine-learning problems.

### 3.3.4 Different measures afford different inferences

In the points above, we have outlined different attributes that a measure of alignment may have. But which measure should we use? Rather than advocating for a particular measure, our goal is to communicate that different measures are sensitive to different features, and therefore afford different inferences. Indeed, we have been less strict in our analysis than some prior works (e.g. Williams et al., 2021); for example, we do not require that a measure satisfy the mathematical criteria of a metric (e.g. we accept asymmetric measures). However, these distinct features can each be advantageous in certain situations.

As a simple conceptual example, suppose that one system encodes signal A in 99% of its neurons and signal B in 1%, whereas another encodes signal A in 1% and signal B in 99%. Regression would quantify these systems as identical, despite most of their activity serving different purposes. RSA would classify them as very dissimilar, despite them representing exactly the same information.

| Alignment function ($\delta$) | (Dis-)Similarity | Descriptive/Differentiable | Symmetric/Directional |
|---|---|---|---|
| Centered Kernel Alignment (CKA) | Similarity | Descriptive & differentiable | $\delta(x,y) = \delta(y,x)$ |
| Pearson RDM correlation | Similarity | Descriptive & differentiable | $\delta(x,y) = \delta(y,x)$ |
| RDM rank correlation coefficient $\rho_a$ | Similarity | Descriptive | $\delta(x,y) = \delta(y,x)$ |
| whitened unbiased RDM cos-similarity | Similarity | Descriptive & differentiable | $\delta(x,y) = \delta(y,x)$ |
| RDM cos-similarity | Similarity | Differentiable | $\delta(x,y) = \delta(y,x)$ |
| Mutual Information (MI) | Similarity | Descriptive | $\delta(x,y) = \delta(y,x)$ |
| $\ell_2$-distance | Dissimilarity | Differentiable | $\delta(x,y) = \delta(y,x)$ |
| KL-divergence (KL) | Dissimilarity | Differentiable | $\delta(x,y) \neq \delta(y,x)$ |
| Cross-entropy (CE) | Dissimilarity | Differentiable | $\delta(x,y) \neq \delta(y,x)$ |

Table 1: Examples of alignment functions and their properties.

More generally, symmetric measures of alignment can be more intuitive, but also elide important distinctions, such as which of two systems contains more information, or which is noisier (though see Duong et al. 2023). Asymmetric measures can provide more insight into features like these, but can lead to other kinds of failures (as above). Likewise, measures that do not fit parameters may underestimate how similar two systems are, if they use slightly different coding schemes that capture on the same information, However, sometimes methods that fit parameters — even using methods as simple as linear regression — can be too flexible (Conwell et al., 2022).

There is also a question of how to normalize measures; e.g. many analyses require specifying a notion of maximum-achievable alignment. For example, in the presence of noise, this is often denoted by the "noise ceiling" estimated by comparing representational predictivity across subsets of the data (e.g. Yamins et al., 2014)—the representational alignment with another system would generally not be expected to exceed this threshold.[4] A more sophisticated method is proposed by (Thobani et al., 2025), who use inter-subject transforms to effectively normalize a measure of model-subject similarity.

Depending on the measures (and normalizations) we use, we may arrive at very different conclusions. Thus, where possible, it is useful to consider multiple measures of similarity and evaluate how conclusions generalize (see §5.3 for further discussion). Alignment measures are an active area of research, including work on measures that more naturally capture relationships between representations that incorporate unit-level tuning without being restricted to it (Khosla and Williams, 2023), measures that can be reliable over small datasets (Pospisil et al., 2024), and frameworks that bridge between or unify different measures (Harvey et al., 2023; 2024; Williams, 2024).

### 3.3.5 What does it take to unambiguously specify a similarity measure?

Comparing similarity scores across studies can be challenging due to variability in naming and implementation conventions (Cloos et al., 2024b). As an illustrative example, consider the similarity measure CKA. CKA was defined by Kornblith et al. (2019) in terms of a quantity called the Hilbert-Schmidt Independence Criterion (HSIC).

$$\text{CKA}(\boldsymbol{K}, \boldsymbol{L}) = \frac{\text{HSIC}(\boldsymbol{K}, \boldsymbol{L})}{\sqrt{\text{HSIC}(\boldsymbol{K}, \boldsymbol{K})\text{HSIC}(\boldsymbol{L}, \boldsymbol{L})}}$$

CKA is most commonly written using the linear kernel $\boldsymbol{K} = \boldsymbol{V}\boldsymbol{V}^\top$, $\boldsymbol{L} = \boldsymbol{W}\boldsymbol{W}^\top$ and the estimator for HSIC originally proposed by Gretton et al. (2005) as in section 4.3.1.

$$\text{HSIC}_{\text{Gretton}}(\boldsymbol{K}, \boldsymbol{L}) = \frac{1}{(n-1)^2} \text{Tr}(\boldsymbol{K}\boldsymbol{H}\boldsymbol{L}\boldsymbol{H})$$

where $\boldsymbol{H}_n = \boldsymbol{I}_n - \frac{1}{n}\boldsymbol{1}\boldsymbol{1}^\top$ is the centering matrix. However, this estimator is biased (Gretton et al., 2005). Subsequently, Song et al. (2007) proposed an unbiased estimator of HSIC.

---

[4]Except in the presence of unaccounted-for confounds, e.g. fixed stimulus orders across subjects.

$$\text{HSIC}_{\text{Song}}(\boldsymbol{K}, \boldsymbol{L}) = \frac{1}{n(n-3)} \left[ \text{Tr}(\tilde{\boldsymbol{K}}\tilde{\boldsymbol{L}}) + \frac{\mathbf{1}^{\top}\tilde{\boldsymbol{K}}\mathbf{1}\,\mathbf{1}^{\top}\tilde{\boldsymbol{L}}\mathbf{1}}{(n-1)(n-2)} - \frac{2}{n-2}\mathbf{1}^{\top}\tilde{\boldsymbol{K}}\tilde{\boldsymbol{L}}\mathbf{1} \right]$$

where $\tilde{\boldsymbol{K}}_{ij} = (1 - \delta_{ij})\boldsymbol{K}_{ij}$ and $\tilde{\boldsymbol{L}}_{ij} = (1 - \delta_{ij})\boldsymbol{L}_{ij}$ are the kernel matrices with diagonal entries set to zero. Additionally, Lange et al. (2023) proposed an estimator that can be both written as an inner product and that has low bias.

$$\text{HSIC}_{\text{Lange}}(\boldsymbol{K}, \boldsymbol{L}) = \frac{2}{n(n-3)} \left\langle \text{tril}(\boldsymbol{H}\boldsymbol{K}\boldsymbol{H}), \text{tril}(\boldsymbol{H}\boldsymbol{L}\boldsymbol{H}) \right\rangle_F$$

where $\text{tril}(\boldsymbol{A})$ denotes the vector formed by the elements of the lower triangular part of matrix $\boldsymbol{A}$, excluding the diagonal, and $\langle \cdot, \cdot \rangle_F$ denotes the Frobenius inner product.

These choices for the HSIC estimator are not just subtleties of the implementation but can have a large impact on the final similarity score. Murphy et al. (2024) compared CKA with $\text{HSIC}_{\text{Gretton}}$ to CKA with $\text{HSIC}_{\text{Song}}$ and found that the unbiased estimator $\text{HSIC}_{\text{Song}}$ is better at detecting stimuli-driven alignment in fMRI and MEG data.

As this example shows, in order to more unambiguously identify a particular implementation of CKA we would need to also specify how HSIC is estimated. Additionally, there are other variations to specify, for example, Williams et al. (2021) proposed taking the arccosine of CKA to satisfy the axioms of a metric; Ding et al. (2021) used 1- CKA; Huh et al. (2024) used a local version of CKA that considers only the top-k nearest neighbors. To facilitate comparisons across different studies and make explicit the implementation choices underlying a given code repository Cloos et al. (2024a) created, and are continuing to develop, a Python package that benchmarks and standardizes similarity measures. The goal of this repository is to gather existing implementations of similarity measures with a common naming convention and customizable interface, ultimately making it easier for the community to make comparisons across studies.

## 4 Universal notation across diverse communities

The goal of our framework is to introduce a common language that can highlight similarities in the approaches and goals across a diversity of fields concerned with the alignment of intelligent systems. To demonstrate how our framework fulfills this role, in this section, we describe how representational alignment plays a role in specific research projects (those visualized in Figure 1). For each of the highlighted examples, we present a short conceptual summary followed by a formal mathematical description structured according to the framework. We hope that illustrating *how* alignment is studied by different communities will enable readers to see connections between topics, and hopefully empower them to transfer best practices from other research communities to their own research topics. Table 2 provides a concise summary of how additional related literature fits into the unifying framework.

### 4.1 Cognitive Science

#### 4.1.1 Measuring representational alignment (Figure 1a)

Jacoby and McDermott (2017) use a serial reproduction paradigm (Griffiths and Kalish, 2005) to elicit rhythm priors from participants. In this paradigm participants are initially introduced with a simple rhythm that is randomized from the possible "universe"' of simple rhythmic patterns. Participants reproduced the pattern, and the average reproduction became the stimulus for a new iteration. After repeating the process a fixed number of times, the experimenter identifies the density of responses within the stimulus space. In this way, categories emerge as high-density response areas. One can show that this paradigm, under certain experimentally verifiable conditions, converges to a sample from the perceptual prior over the relevant domain (Griffiths and Kalish, 2005; Langlois et al., 2021b). Jacoby and McDermott (2017) showed that categories identified with this method overlap with integer ratios, and that they differ between speech and musical stimuli. A big advantage of this paradigm is that it can be used to study non-experts and participants with no musical experience as it relies on minimal verbal instructions. A large-scale cross-cultural replication of this work (Jacoby et al., 2021a) tested the paradigm with 39 groups from 15 countries. The results showed categorical prototypes in all cultures that are near simple integer ratios. However, the weight (importance) of categories varied substantially from place to place. This is in contrast to another follow-up work where

| | Data | Systems | | Alignment | |
|---|---|---|---|---|---|
| Research paper(s) \ Setting | Trials | A | B | Objective | $\delta(x, y)$ |
| 210; 285 | Images | Monkey (brain) | Human (brain) | measuring | RDM rank correlation |
| 198; 135; 40; 41; 395 | Images | Human (brain) | Human (behavior) | measuring | RDM rank correlation |
| 285; 183; 135 | Images | Human (brain) | Human (behavior) | measuring | RDM linear combination |
| 393 | Images | Mouse (brain) | Mouse (brain) | measuring | RDM optimal transport |
| 191 | Images | Monkey (brain) | DNN | measuring | RDM rank correlation |
| 5; 67; 435; 76; 308 | Images | Human (brain) | DNN | measuring | RDM rank correlation |
| 95; 195; 191 | Images | Human (brain) | DNN | measuring | RDM linear combination |
| 383; 288; 289; 321; 256; 257; 212; 159; 158 | Images | Human (behavior) | DNN | measuring | RDM rank correlation |
| 198; 135; 40; 41; 395 | Images | Human (brain, behavior) | DNN | measuring | RDM rank correlation |
| 286 | Images | Human (brain) | DNN | measuring | CKA |
| 392 | Images | Human (behavior) | DNN | measuring | RDM optimal transport |
| 300; 38 | Images | Human (brain, behavior) | DNN | measuring | Task accuracy |
| 383 | Images | Human (behavior) | DNN | measuring | Pearson RDM correlation |
| 390; 283 | Images | Human (behavior) | DNN | measuring | Procrustes measure |
| 233 | Images, Video | Human (behavior) | DNN | measuring | Euclidean distance |
| 260; 257 | Audio, Video | Human (behavior) | DNN | measuring | Pearson RDM correlation |
| 86; 87 | Video | Human (brain) | DNN | measuring | RDM rank correlation |
| 406; 87 | Text | Human (brain) | DNN | measuring | RDM rank correlation |
| 20 | Text | Human (behavior) | Human (behavior) | measuring | RDM rank correlation |
| 256; 260; 257 | Text | Human (behavior) | LLM | measuring | Pearson RDM correlation |
| 237; 239 | Text | Human (behavior) | LLM | measuring | KL-divergence |
| 389 | Text | Human (behavior) | LLM, multi-modal | measuring | Procrustes measure |
| 394 | Odorants | Human (behavior) | LLM, multi-modal | measuring | Pearson RDM correlation |
| 189 | Colors | Human (behavior) | Human (behavior) | measuring | RDM optimal transport |
| 205; 45 | Images | DNN | DNN | measuring | CKA |
| 243; 155 | Images | DNN | DNN | measuring | Pearson RDM correlation |
| 281 | Images | DNN | DNN | measuring | RDM cos-similarity |
| 310 | Time series | RNN | RNN | measuring | Angular Procrustes |
| 399 | Text | LLM | LLM | measuring | Pearson correlation |
| 437; 295; 49; 50 | Images | Monkey (brain) | DNN | bridging | $\ell_2$-distance |
| 69 | Images | Monkey (brain) | RNN | bridging | $\ell_2$-distance, CKA |
| 367 | Images | Monkey (brain), Human (brain) | DNN | bridging | Task accuracy |
| 329 | Images | Mouse (brain) | DNN | bridging | cosine distance |
| 192; 191; 76; 382; 136; 203; 378; 194 | Images | Human (brain) | DNN | bridging | $\ell_2$-distance |
| 327 | Phosphenes | Human (brain) | DNN | bridging | $\ell_2$-distance |
| 297; 419; 402 | Images, Text | Human (brain) | DNN, LLM, multi-modal | bridging | $\ell_2$-distance |
| 395 | Images | Human (brain, behavior) | DNN | bridging | RDM rank correlation |
| 194 | Images | Human (brain) | DNN | bridging | Pearson correlation |
| 381 | Images | Human (behavior) | DNN | bridging | RMSE |
| 405 | Images | Human (behavior) | DNN | bridging | Pearson RDM correlation |
| 238 | Images | Human (behavior) | DNN | bridging | Cross-entropy |
| 207 | Images | RL agent | RL agent | bridging | RDM rank correlation |
| 236 | Images | Human (behavior) | Diffusion model | bridging | Cosine similarity |
| 265; 137; 219; 116 | Video | Human (brain) | DNN | bridging | $\ell_2$-distance |
| 361; 13; 14 | Text | Human (brain) | LLM | bridging | $\ell_2$-distance |
| 425; 239; 20 | Text | Human (behavior) | LLM | bridging | $\ell_2$-distance |
| 124 | Images | DNN | DNN | bridging | CKA |
| 103 | Images | Monkey (brain) | DNN | increasing | RDM cos-similarity |
| 80 | Images | Monkey (brain) | DNN | increasing | CKA |
| 288; 289 | Images | Human (behavior) | DNN | increasing | Cross-entropy |
| 73; 72; 385 | Images | Human (behavior) | DNN | increasing | $\ell_2$-distance |
| 112; 388 | Images | Human (behavior) | DNN | increasing | Hinge Loss |
| 290 | Images | Human (behavior) | DNN | increasing | KL-divergence |
| 167 | Images | DNN | DNN | increasing | Cross-entropy |
| 160; 62; 410; 323; 384 | Images | DNN | DNN | increasing | KL-divergence |
| 263 | Images | DNN | DNN | increasing | CCA |
| 422 | Images | DNN | DNN | increasing | Procrustes measure |
| 244 | Text | LLM | LLM | increasing | $\ell_2$-distance |

Table 2: Examples of research articles from cognitive science, neuroscience, machine learning, and other fields, that relate to representational alignment. This table is intended to illustrate the broad interdisciplinary nature of the field of representational alignment, rather than to provide a complete overview of the literature. It explicitly features studies that were accepted to the `Re-Align` workshop at ICLR (Grant et al., 2024). We encourage readers to send us suggestions for making this table more comprehensive.

American and Canadian children were tested (Nave et al., 2024). Here, there were small differences between adults and children underscoring the idea that rhythm presentations are learned at an early age.

- **Data** $\mathcal{D}$**:** Let $\mathcal{D} := \{(i_1, i_2, i_3) \mid i_1 + i_2 + i_3 = T, \min(i1, i2, i3) > f\}$ be all possible 3-interval rhythms, where $T$ is the total duration, $i_1$, $i_2$, and $i_3$ are the three intervals and $f$ is the minimal possible interval (so that we avoid presenting rhythms that are too short).

- **System** A: Let $f_\theta$ be a representative group of human subjects who perform the task. The analysis is done at the group level and the output is a probability function (kernel density) of the three-interval space.

  - **Its measurements $X$:** Human tapping response for $n$ randomly sampled initial seeds from $\mathcal{D}$. Data was collected from a group of $m$ participants. Participants perform the serial reproduction process and repeat the initial seed. The seed becomes the input of new iterations. After a finite number of iterations (typically $K = 5$) the process stops and a new block begins with another random seed.
  - **Its embedding $V$:** $V$ is the kernel density estimate for the data from the last two iterations.

- **System** B: This function stems from the same system as the function $f_\theta$ but for another group of people — hence, $g_\phi$ — with corresponding measurements $Y$ and embeddings $W$. For example, the first system can be participants from the US and the second system can be participants from the Bolivian Amazon.

- **Differentiable and symmetric alignment function** $\delta(V, W)$: $\mathrm{JSD}(V\|W) = \frac{1}{2}\sum V(\log V - \log M) + \frac{1}{2}\sum W(\log W - \log M)$ is the Jensen–Shannon divergence computed over the two kernel density functions where $M = \frac{1}{2}(V + W)$ is a mixture distribution of the two kernels.

### 4.1.2 Bridging representational spaces (Figure 1d)

Hebart et al. (2020) collected 1.46 million human triplet odd-one-out judgments to generate a sparse positive similarity embedding (SPOSE; Zheng et al., 2019) underlying these similarity judgments. In contrast to much previous work that has manually identified candidate dimensions, focused on small, non-representative representational spaces, or yielded low interpretability, Hebart et al. (2020) revealed 49 interpretable embedding dimensions in a data-driven fashion for a broad set of 1854 object categories that were highly predictive of single trial choice behavior. Instead of comparing representations using representational similarity analysis (Kriegeskorte et al., 2008a) or similar measures, this approach of identifying core representational dimensions allows for direct comparison of candidate dimensions that determine representational alignment. Therefore, it provides a pathway for interpretable representational alignment between different individuals or modalities.

- **Data $\mathcal{D}$:** Let $\mathcal{D} := (\{i_s, j_s, k_s\})_{s=1}^n$ be a dataset of $n$ sets of three objects where each object in the triad is an image. Let $m$ denote the number of distinct objects in this dataset where $m = 1854$.

- **System** A: Let $f_\theta$ be a representative human participant who outputs a discrete (odd-one-out) choice for each triplet in the data. The analysis is done at the participant level with choices pooled across participants, and the output is an odd-one-out choice for each triplet in the data.

  - **Its measurements $X$:** Asking each human participant to select the odd-one-out object for each triplet in the data yields $X := (\{a_s, b_s\} \mid \{i_s, j_s, k_s\})_{s=1}^n$, a human-response dataset of $n$ ordered tuples of discrete choices. Note that $f_\theta$ is a non-deterministic function and thus its measurements are sampled from different human participants (the responses might as well be aggregated).
  - **Its embedding $V$:** Let $\varphi_\psi(x)$ be a differentiable embedding function with learnable variables $\boldsymbol{W}_X \in \mathbb{R}^{m \times p}$ where $p \ll m$ and $\boldsymbol{W}$ is initialized with Gaussian random variables. Let $S_{ij} := \boldsymbol{w}_i^\top \boldsymbol{w}_j$ indicate the similarity between object representations $\boldsymbol{w}_i, \boldsymbol{w}_j$ in the $p$-dimensional embedding space where $S_X \in \mathbb{R}^{m \times m}$ is the affinity matrix of all pairwise object similarities. Thus, the embedding $V := W$ is the learnable variables.

- **System** B: There is no function $g_\phi$ in Hebart et al. (2020) but in principle one can imagine this function to stem from the same system as the function $f_\phi$ but for another group of people (e.g., different cultural groups). However, it might as well stem from other systems, such as neural network representations or brain data. In the latter cases, no triplets are directly accessible, but we can easily generate them from the measurements of the function $g_\phi$, where the measurements

$\boldsymbol{Y} \coloneqq (g_\phi(s_1), \ldots, g_\phi(s_m)) \in \mathbb{R}^{m \times d}$ are a stacked matrix of $m$ object representations[5] from which we can infer a similarity matrix (e.g., $\boldsymbol{S}_Y \coloneqq \boldsymbol{Y}\boldsymbol{Y}^\top$). Subsequently, we can sample triplets from $S_Y$ and learn the low-dimensional SPoSE embedding using these generated triplets.

- **Differentiable and directional alignment function** $\delta(X, \boldsymbol{W})$**:**

$$\delta(X, \boldsymbol{W}) \coloneqq \arg\min_W \frac{1}{n} - \sum_{s=1}^{n} \log p(\{a_s, b_s\} \mid \{i_s, j_s, k_s\}, \boldsymbol{W}) + \lambda \|\boldsymbol{W}\|_1,$$

where $p(\{a_s, b_s\} \mid \{i_s, j_s, k_s\}, \boldsymbol{W}) = \exp\left(\boldsymbol{w}_a^\top \boldsymbol{w}_b\right) / \left(\exp\left(\boldsymbol{w}_i^\top \boldsymbol{w}_j\right) + \exp\left(\boldsymbol{w}_i^\top \boldsymbol{w}_k\right) + \exp\left(\boldsymbol{w}_j^\top \boldsymbol{w}_k\right)\right)$ and $\lambda$ is a hyper-parameter that determines the strength of the sparsity-inducing $\ell_1$-regularization.

Similarly, Muttenthaler et al. (2022) used the same set of measurements in combination with a similar alignment function (same data log-likelihood function but different regularization) for learning a more robust version of the embedding $\boldsymbol{W}$ using approximate Bayesian inference. They used a *spike-and-slab* Gaussian mixture prior instead of vanilla $\ell_1$-regularization and learned a matrix for the variance over the human odd-one-out choices in addition to the (mean) embedding matrix, demonstrating that this more appropriate than the above deterministic version when $n$ is small.

### 4.1.3 Increasing representational alignment (Figure 1g)

Muttenthaler et al. (2023b) use human triplet odd-one-out choices to increase the alignment between neural network representation and human object similarity spaces. The human odd-one-out choices were collected using large-scale online crowd-sourcing in a previous study (Hebart et al., 2020). The objective in Muttenthaler et al. (2023b) was to align a neural network function $f_\theta$ with the behavior of human participants $g_\phi$ where $g_\phi$ is not a deterministic function and, thus, the human behavior is aggregated across multiple participants. That is, their goal was to perform *representational transformation* (see §3.3.2) from the neural network representation space into the human object similarity space. Therefore, they used a *directional* and *differentiable* alignment function which — as we have seen in §3.3 — are both desirable but not necessary properties of an alignment function.

- **Data** $\mathcal{D}$: Let $\mathcal{D} \coloneqq (\{i_s, j_s, k_s\})_{s=1}^{n}$ be a dataset of $n$ sets of three objects where each object in the triplet is an image. Let $m$ denote the number of distinct objects in this dataset where $m = 1854$.

- **System** A: Let $f_\theta : \mathbb{R}^{H \times W \times C} \mapsto \mathbb{R}^p$ be a deterministic neural network function parametrized by $\theta$ that maps an image tensor to a $p$-dimensional vector representation (in its penultimate layer/image encoder space).

  - **Its measurements** $X$: Applying $f_\theta$ to each image in the data yields $\boldsymbol{X} \coloneqq (f_\theta(s_1), \ldots, f_\theta(s_m)) \in \mathbb{R}^{m \times p}$, a stacked matrix of $m$ (penultimate layer) object representations.
  - **Its embedding** $V$: Let $S_{ij} \coloneqq \boldsymbol{x}_i^\top \boldsymbol{x}_j$ be the similarity between object representations $\boldsymbol{x}_i, \boldsymbol{x}_j$ in the original representation space and $V_{ij} = \varphi_\psi(X_{ij}) \coloneqq (\boldsymbol{W}\boldsymbol{x}_i + \boldsymbol{b})^\top (\boldsymbol{W}\boldsymbol{x}_j + \boldsymbol{b})$ indicate the similarity between object representations $\boldsymbol{x}_i, \boldsymbol{x}_j$ in the transformed representation space. So, $\boldsymbol{V} \in \mathbb{R}^{m \times m}$ is the affinity matrix of all pairwise object similarities in the transformed space. Here, the transformation matrix $\boldsymbol{W} \in \mathbb{R}^{p \times p}$ and the bias vector $\boldsymbol{b} \in \mathbb{R}^p$ are both learnable variables (optimized via SGD).

- **System** B: Let $g_\phi$ be a representative human participant who outputs a discrete (odd-one-out) choice for each triplet in the data.

  - **Its measurements** $Y$: Asking each human participant to select the odd-one-out object for each triplet in the data yields $Y \coloneqq (\{a_s, b_s\} \mid \{i_s, j_s, k_s\})_{s=1}^{n}$, a human-response dataset of $n$ ordered tuples of discrete choices. Note that $g_\phi$ is a non-deterministic function and thus its measurements are sampled from different human participants.

---

[5]The dimensionality $d$ of the object representations may or may not be collapsed. It may be collapsed if the representations are inferred from brain data or from a convolutional layer of a CNN which are both generally of tensor format.

– **Its embedding** $W$: There is no embedding function. Here, $W = Y$, a human response dataset of discrete odd-one-out choices.

- **Differentiable and directional alignment function** $\delta(V, W)$:

$$\delta(V, W) := \operatorname*{arg\,min}_{\boldsymbol{W}, \boldsymbol{b}} \frac{1}{n} - \sum_{s=1}^{n} \log p(\{a_s, b_s\} \mid \{i_s, j_s, k_s\}, V) + \lambda \left\| \boldsymbol{W} - \left( \sum_{j=1}^{p} \boldsymbol{W}_{jj}/p \right) \boldsymbol{I} \right\|_{\mathrm{F}}^2,$$

where $p(\{a_s, b_s\} \mid \{i_s, j_s, k_s\}, \boldsymbol{V}) = \exp\left(\boldsymbol{v}_a^\top \boldsymbol{v}_b\right) / \left(\exp\left(\boldsymbol{v}_i^\top \boldsymbol{v}_j\right) + \exp\left(\boldsymbol{v}_i^\top \boldsymbol{v}_k\right) + \exp\left(\boldsymbol{v}_j^\top \boldsymbol{v}_k\right)\right)$ and $\lambda$ is a hyper-parameter that determines the strength of the $\ell_2$-regularization.

Using the above (constrained) alignment function plus an additional contrastive learning objective that preserves the local similarity structure from the original neural network representation space allowed the authors to obtain a human-aligned representation space that showed increased representational alignment with human perception and better downstream task performance on various computer vision tasks (Muttenthaler et al., 2023b).

### 4.2 Neuroscience

#### 4.2.1 Measuring representational alignment (Figure 1b)

Kriegeskorte et al. (2008b) used RSA to measure alignment between neural responses in monkey and human inferotemporal cortex. The monkey neural responses were measured with multi-array electrophysiology and the human neural responses were measured with fMRI. The objective was to compare the representational geometry across monkeys and humans to determine if IT cortex is homologous across primate species using a descriptive and symmetric alignment function.

- **Dataset** $\mathcal{D}$: Let $\mathcal{D} := \{s_i\}_{i=1}^n$ be a set of $n$ images depicting objects on plain white backgrounds.

- **System** A: Let $f_\theta : \mathbb{R}^{H \times W \times C} \mapsto \mathbb{R}^p$ be a Rhesus macaque monkey whose neural activity we want to record for each image in the data using electrophysiology measures. A monkey is a non-deterministic function parametrized by $\theta$.

  – **Its measurements** $X$: Let $\boldsymbol{X} := (f_\theta(s_1), \ldots, f_\theta(s_n)) \in \mathbb{R}^{n \times p}$ be the stacked monkey's electrophysiology signals from inferior temporal cortex for each image in the data $\mathcal{D}$. For each image, the electrophysiology measurements are represented by a vector of $p$ electrodes that reflect neural activity.

  – **Its embedding** $V$: Upper-triangular off-diagonal elements of the representational dissimilarity matrix $\boldsymbol{S}_X \in \mathbb{R}^{n \times n}$ where each entry $s_{ij}^X := 1 - \left( (\boldsymbol{x}_i - \bar{\boldsymbol{x}}_i)^\top (\boldsymbol{x}_j - \bar{\boldsymbol{x}}_j) / \left( \|\boldsymbol{x}_i - \bar{\boldsymbol{x}}_i\|_2 \|\boldsymbol{x}_j - \bar{\boldsymbol{x}}_j\|_2 \right) \right)$ is determined by 1 minus the Pearson correlation coefficient between image representations $\boldsymbol{x}_i, \boldsymbol{x}_j$. Thus, we have that the embedding $\boldsymbol{v} \in \mathbb{R}^{nn/2-n}$ is a (flattened) vector representation rather than a matrix.

- **System** B: Let $g_\phi : \mathbb{R}^{H \times W \times C} \mapsto \mathbb{R}^{v \times d}$ be a human participant who transforms images into neural activity. A human participant is a non-deterministic function parametrized by $\phi$.

  – **Its measurements** $Y$: Let $\boldsymbol{Y} := (g_\phi(s_1), \ldots, g_\phi(s_n)) \in \mathbb{R}^{n \times v \times d}$ be the human participant's fMRI responses from inferior temporal cortex for each image in the data $\mathcal{D}$. For each image, the fMRI responses are represented by a matrix of voxel $\times$ individual neuron activities with $v$ voxels and $d$ neurons.

  – **Its embedding** $W$: Upper-triangular off-diagonal elements of the representational dissimilarity matrix $\boldsymbol{S}_Y \in \mathbb{R}^{n \times n}$ where each entry $s_{ij}^Y := 1 - \left( (\boldsymbol{y}_i - \bar{\boldsymbol{y}}_i)^\top (\boldsymbol{y}_j - \bar{\boldsymbol{y}}_j) / \left( \|\boldsymbol{y}_i - \bar{\boldsymbol{y}}_i\|_2 \|\boldsymbol{y}_j - \bar{\boldsymbol{y}}_j\|_2 \right) \right)$ is determined by 1 minus the Pearson correlation coefficient between image representations $\boldsymbol{y}_i, \boldsymbol{y}_j$. Thus, we have that the embedding $\boldsymbol{w} \in \mathbb{R}^{nn/2-n}$ is a (flattened) vector representation of the same shape as $\boldsymbol{v}$.

- **Descriptive and symmetric alignment function** $\delta(\boldsymbol{v}, \boldsymbol{w})$**:** Spearman's rank correlation coefficient between the embedding vectors $\boldsymbol{v}$ and $\boldsymbol{w}$. Note that the Spearman rank correlation is non-differentiable.

### 4.2.2 Bridging representational spaces (Figure 1e)

O'Connell and Chun (2018) introduced techniques to (a) align fMRI responses across different individuals and (b) align fMRI responses to eye movement behavior within individuals. Humans viewed images depicting natural scenes while undergoing fMRI scanning, then in a separate session viewed the images while their eye movements were recorded. To align brain activity across individuals, a linear decoding analysis was used to map each individual's fMRI responses into a common space defined as the unit activity of a CNN, which allowed for group-level analysis over the mean of the aligned responses. To align human brain activity to eye movements, a computational salience model is applied to the CNN-aligned fMRI responses to derive a brain-based spatial priority map which was then compared to human eye movement patterns. The objective was to identify brain regions in humans that capture spatial information predictive of human eye movement patterns.

(a) *aligning fMRI responses across individuals into a common (CNN-determined) representation space*:

- **Data** $\mathcal{D}$**:** Let $\mathcal{D} \coloneqq \{s_i\}_{i=1}^n$ be a set of $n$ images, each depicting a natural scene.
- **System** A: Let $f_\theta : \mathbb{R}^{H \times W \times C} \mapsto \mathbb{R}^{v \times p}$ be a human participant who transforms images into neural activity. A human participant is a non-deterministic function parametrized by $\theta$.
  - **Its measurements** $X$**:** Let $\boldsymbol{X} \coloneqq (f_\theta(s_1), \ldots, f_\theta(s_n)) \in \mathbb{R}^{n \times v \times p}$ be the stacked individual's fMRI responses for each image in the data $\mathcal{D}$. For each image, the fMRI responses are represented by a matrix of voxel $\times$ individual neuron activities with $v$ voxels and $p$ neurons.
  - **Its embedding** $V$**:** Let $\varphi_\psi(\boldsymbol{x}_i) : \mathbb{R}^{v \times p} \mapsto \mathbb{R}^d$ denote partial least-squares (PLS) regression that learns a linear transformation from the participant's measurements space $X$ to the representation space of a CNN. The transformation was applied to held-out data to map the individual fMRI responses to the embedding space such that $\boldsymbol{V} \coloneqq (\varphi_\psi(\boldsymbol{x}_1), \ldots, \varphi_\psi(\boldsymbol{x}_n)) \in \mathbb{R}^{n \times d}$. Note that a flattening operation was applied to the rows of $\boldsymbol{X}$ before employing PLS regression.
- **System** B: Let $g_\phi : \mathbb{R}^{H \times W \times C} \mapsto \mathbb{R}^{v \times p}$ be a different human participant who transforms images into neural activity.
  - **Its measurements** $Y$**:** Let $\boldsymbol{Y} \coloneqq (g_\phi(s_1), \ldots, g_\phi(s_n)) \in \mathbb{R}^{n \times v \times p}$ be the individual's fMRI responses for each natural scenes image in the data $\mathcal{D}$.
  - **Its embedding** $W$**:** The same PLS regression mapping as above was used to map from the participant's measurements space $\boldsymbol{Y}$ to the representation space of a CNN. Similarly, the transformation was applied to held-out data to map the individual fMRI responses to the embedding space such that $\boldsymbol{W} \coloneqq (\chi_\omega(\boldsymbol{y}_1), \ldots, \chi_\omega(\boldsymbol{y}_n)) \in \mathbb{R}^{n \times d}$.
- **Symmetric alignment function** $\delta(V, W)$**:** $\delta(\boldsymbol{v}_i, \boldsymbol{w}_i) \coloneqq \frac{(\boldsymbol{v}_i - \bar{\boldsymbol{v}}_i)^\top (\boldsymbol{w}_i - \bar{\boldsymbol{w}}_i)}{\|\boldsymbol{v}_i - \bar{\boldsymbol{v}}_i\|_2 \|\boldsymbol{w}_i - \bar{\boldsymbol{w}}_i\|_2}$, where $\delta(\boldsymbol{v}_i, \boldsymbol{w}_i)$ denotes the Pearson correlation (coefficient) between the representations of function $f_\theta$ and function $g_\phi$ respectively for the same image in the shared (CNN-determined) embedding space.

(b) *aligning fMRI responses to eye movement behavior*:

- **Data** $\mathcal{D}$**:** Let $\mathcal{D} \coloneqq \{s_i\}_{i=1}^n$ be the same set of images as in (a).
- **System** A: Let $f_\theta : \mathbb{R}^{H \times W \times C} \mapsto \mathbb{R}^{v \times p}$ be a human participant whose neural activity is recorded for each image in the data.
  - **Its measurements** $X$**:** Let $\boldsymbol{X} \coloneqq (\varphi_\psi(f_\theta(s_1)), \ldots, \varphi_\psi(f_\theta(s_n))) \in \mathbb{R}^{n \times d}$ be the stacked group-level human fMRI responses transformed into a shared (CNN-determined) representation space (see embedding space above).
  - **Its embedding** $V$**:** The group-level CNN-transformed fMRI responses were averaged across the CNN activity feature dimension and layers to derive a brain-based spatial priority map predicting where people would look in an image. So, $\boldsymbol{V} \in \mathbb{R}^{n \times m}$

- **System** B: Let $g_\phi : \mathbb{R}^{H \times W \times C} \mapsto \mathbb{R}^{t \times z}$ be the same human participant whose continuous eye movement patterns (instead of neural activity) is recorded for each image in the data.
  - **Its measurements $Y$:** Let $\boldsymbol{Y} := (g_\phi(s_1), \ldots, g_\phi(s_n)) \in \mathbb{R}^{n \times t \times z}$ be the individual participant's (continuous)) eye movement recordings (derived from an eye-tracking camera) for each image in the data $\mathcal{D}$.
  - **Its embedding $W$:** Let $W = \{x_i, y_i\}_{i=1}^n$ be the set of $(x, y) \in \mathbb{R}_+^2$ coordinates defining the location of all fixations for a given image in $\mathcal{D}$ where $n$ is the number of fixations.
- **Descrtiptive and directional alignment function $\delta(V, W)$:** The Normalized Scanpath Salience (NSS) is the mean of the spatial priority map activations corresponding to fixation locations such that $\mathrm{NSS}(\boldsymbol{v}, \boldsymbol{w}) = \frac{1}{n} \sum_{(a,b) \in W} V_{ab}$.

### 4.2.3 Increasing representational alignment (Figure 1h)

Khosla and Wehbe (2022) trained CNNs to predict human fMRI responses in visual brain regions. While previous work had compared alignment in fMRI and image-optimized CNN representations using descriptive measures, this work aimed to increase human fMRI and CNN alignment by directly optimizing CNNs to be aligned with fMRI responses. They find that CNNs optimized to predict responses in high-level visual brain regions recapitulate visual behaviors including classification and making aligned similarity judgments to humans.

- **Data $\mathcal{D}$:** Let $\mathcal{D} := \{s_i\}_{i=1}^n$ be a set of $n$ images, each depicting a natural scene.

- **System** A: Let $f_\theta : \mathbb{R}^{H \times W \times C} \mapsto \mathbb{R}^{v \times p}$ be a human participant whose neural activity we want to measure for a set of images. A human participant is a non-deterministic function parametrized by $\theta$.

  - **Its measurements $X$:** Let $\boldsymbol{X} := (f_\theta(s_1), \ldots, f_\theta(s_n)) \in \mathbb{R}^{n \times v \times p}$ be the individual's fMRI responses for each image in the data $\mathcal{D}$. For each image, the fMRI responses are represented by a matrix of voxel $\times$ individual neuron activities with $v$ voxels and $p$ neurons.
  - **Its embedding $V$:** Let $\varphi_\psi : \mathbb{R}^{v \times p} \mapsto \mathbb{R}^v$ be an aggregation function that maps a matrix of voxel by neuron activities to a single activity per voxel. Thus, $\boldsymbol{V} \in \mathbb{R}^{n \times v}$.

- **System** B: Let $g_\phi : \mathbb{R}^{H \times W \times C} \mapsto \mathbb{R}^d$ be a deterministic neural network function parametrized by $\phi$ that maps an image tensor to a $d$-dimensional vector representation (in its penultimate layer space).

  - **Its measurements $Y$:** Applying $g_\phi$ to each image in the data $\mathcal{D}$ yields $\boldsymbol{Y} := (g_\phi(s_1), \ldots, g_\phi(s_n)) \in \mathbb{R}^{n \times d}$, a stacked matrix of $n$ (penultimate layer) image representations.
  - **Its embedding $W$:** Let $\chi_\omega : \mathbb{R}^d \mapsto \mathbb{R}^v$ be a factorized linear readout (with learnable variables) that transforms penultimate layer image representations into human brain activity. Therefore, $\boldsymbol{W} := (\chi_\omega(\boldsymbol{y}_1), \ldots, \chi_\omega(\boldsymbol{y}_n)) \in \mathbb{R}^{n \times v}$.

- **Differentiable and symmetric alignment function $\delta(V, W)$:** $\mathrm{MSE}(\boldsymbol{v}_i, \boldsymbol{w}_i) = \frac{1}{v} \sum_{j=1}^v (\boldsymbol{v}_{ij} - \boldsymbol{w}_{ij})^2$.

## 4.3 Artificial Intelligence and Machine Learning

### 4.3.1 Measuring representational alignment (Figure 1c)

Just as it is possible to measure the similarity between representations of biological neurons, it is also possible to measure the similarity between representations of artificial neural networks. A variety of neural network representational similarity measures have been proposed (Raghu et al., 2017; Morcos et al., 2018; Williams et al., 2021; Ding et al., 2021). Centered Kernel Alignment (CKA) is a particularly simple and widely-used approach for this purpose (Kornblith et al., 2019):

- **Data $\mathcal{D}$:** Let $\mathcal{D} := \{s_i\}_{i=1}^n$ be a dataset of $n$ images (or text sequences).

- **System** A: Any neural network function. Let $f_\theta : \mathbb{R}^{H \times W \times C} \cup \mathbb{R}^{T \times K} \mapsto \mathbb{R}^p$ be a deterministic neural network function parametrized by $\theta$ that maps a set of inputs (image tensors or text sequences) to a set of $p$-dimensional outputs.

  - **Its measurements** $X$: Let $\boldsymbol{X} \in \mathbb{R}^{n \times p}$ be the matrix of stacked activations extracted from a layer/module of the neural network function $f_\theta$ where $\boldsymbol{X} \coloneqq (f_\theta(s_1), \ldots, f_\theta(s_n))$.

  - **Its embedding** $V$: Here, $\varphi_\psi$ is the identity function (in the case of linear CKA) or an arbitrary feature mapping applied to the set of measurements $\boldsymbol{X}$ where $\boldsymbol{V} \coloneqq (\varphi_\psi(\boldsymbol{x}_1), \ldots, \varphi_\psi(\boldsymbol{x}_n)) \in \mathbb{R}^{n \times m}$.

- **System** B: Any neural network function. Let $g_\phi : \mathbb{R}^{H \times W \times C} \cup \mathbb{R}^{T \times K} \mapsto \mathbb{R}^d$ be another deterministic neural network function parametrized by $\phi$ that maps a set of inputs (image tensors or text sequences) to a set of $d$-dimensional outputs.

  - **Its measurements** $Y$: Let $\boldsymbol{Y} \in \mathbb{R}^{n \times d}$ be the matrix of stacked activations extracted from a layer/module of the neural network function $g_\phi$ where $\boldsymbol{Y} \coloneqq (g_\phi(s_1), \ldots, g_\phi(s_n))$.

  - **Its embedding** $W$: Here, $\chi_\omega$ is the identity function (in the case of linear CKA) or an arbitrary feature mapping applied to the set of measurements $\boldsymbol{Y}$ where $\boldsymbol{W} \coloneqq (\chi_\omega(\boldsymbol{y}_1), \ldots, \chi_\omega(\boldsymbol{y}_n)) \in \mathbb{R}^{n \times z}$.

- **Differentiable and symmetric alignment function** $\delta(V, W)$:

$$\text{CKA} = \frac{\|\boldsymbol{V}^\top \boldsymbol{H} \boldsymbol{W}\|_{\text{F}}^2}{\|\boldsymbol{V}^\top \boldsymbol{H} \boldsymbol{V}\|_{\text{F}} \|\boldsymbol{W}^\top \boldsymbol{H} \boldsymbol{W}\|_{\text{F}}} = \frac{\text{tr}(\boldsymbol{V} \boldsymbol{V}^\top \boldsymbol{H} \boldsymbol{W} \boldsymbol{W}^\top \boldsymbol{H})}{\|\boldsymbol{H} \boldsymbol{V} \boldsymbol{V}^\top \boldsymbol{H}\|_{\text{F}} \|\boldsymbol{H} \boldsymbol{W} \boldsymbol{W}^\top \boldsymbol{H}\|_{\text{F}}},$$

  where $\boldsymbol{H}_n = \boldsymbol{I}_n - \frac{1}{n} \boldsymbol{1} \boldsymbol{1}^\top$ is the centering matrix.

Depending on the choice of the feature mapping, $\boldsymbol{V}$ and $\boldsymbol{W}$ can be expensive or impossible to compute directly. For example, the feature mapping associated with the radial basis function kernel is infinite-dimensional. In these cases one has to compute similarity matrices $\boldsymbol{K} = \boldsymbol{V} \boldsymbol{V}^\top$ and $\boldsymbol{L} = \boldsymbol{W} \boldsymbol{W}^\top$ by evaluating kernel functions $K_{ij} = k(\boldsymbol{x}_i, \boldsymbol{x}_j)$ and $L_{ij} = l(\boldsymbol{x}_i, \boldsymbol{x}_j)$.

### 4.3.2  Bridging representational spaces (Figure 1f)

By enforcing text and image representational alignment, multimodal models achieve better cross-task transfer compared to standard multitask learning. Specifically, Gupta et al. (2017) demonstrate better inductive transfer from visual recognition to visual question answering (VQA) than standard methods, stating that visual recognition additionally improves, in particular for categories that have relatively few recognition training labels but frequently appear in the query setting. Their setup is the following:

- **Data** $\mathcal{D}$: Let $\mathcal{D} \coloneqq \{r_i, w_i\}_{i=1}^n$ be a dataset of $n$ images with corresponding text descriptions.

- **System** A: Any neural network function. Let $f_\theta : \mathbb{R}^{H \times W \times C} \mapsto \mathbb{R}^p$ be a deterministic neural network function parametrized by $\theta$ that maps a set of images to a set of vectorized outputs.

  - **Its measurements** $X$: Let $\boldsymbol{X} \coloneqq (f_\theta(r_1), \ldots, f_\theta(r_n)) \in \mathbb{R}^{n \times p}$ be the stacked average-pooled features from an ImageNet-trained ResNet-50 — which represents the neural network function $f_\theta$ — for all $n$ images in the dataset $\mathcal{D}$.

  - **Its embedding** $V$: Let $\boldsymbol{V} = \boldsymbol{X}$

- **System** B: Any neural network function. Let $g_\phi : \mathbb{R}^{T \times K} \mapsto \mathbb{R}^{t \times d}$ be another deterministic neural network function parametrized by $\phi$ that maps a set of text sequences to a set of vectorized outputs.

  - **Its measurements** $Y$: By applying two fully connected layers (that have 300 output units each) from the neural network function $g_\phi$ to pretrained `word2vec` representations (Mikolov et al., 2013b) of the text descriptions $w$ we obtain $\boldsymbol{Y} \coloneqq (g_\phi(\boldsymbol{w}_1'), \ldots, g_\phi(\boldsymbol{w}_n')) \in \mathbb{R}^{n \times t \times d}$, a stacked tensor of $d$-dimensional representations for each word in the text description.

– **Its embedding** $W$**:** Let $\boldsymbol{W} = \boldsymbol{Y}$

- **Differentiable and directional alignment function** $\delta(V, W)$**:** Depending on whether a word in the text description is an object or an attribute, Gupta et al. (2017) use a different loss function for aligning image and text representations. Therefore, the authors partition the text descriptions into object and attribute sets. If a word in the text description $w_i$ corresponding to an image $r_i$ is an object, then the alignment between the image and text representations is increased by minimizing the following objective,

$$\delta(V, W) := \mathcal{L}_{\mathrm{obj}}\left(f_\theta, g_\phi\right) := \frac{1}{|w_i^{\mathrm{obj}}|} \sum_{l \in w_i^{\mathrm{obj}}}^{|w_i^{\mathrm{obj}}|} \frac{1}{|\mathcal{O}|} \sum_{k \in \{\mathcal{O} \setminus w_i^{\mathrm{obj}}\}} \max\{0, \eta_{\mathrm{obj}} + \boldsymbol{x}_i^\top g_\phi\left(\mathcal{O}\right)_k - \boldsymbol{x}_i^\top Y_{il}^{\mathrm{obj}}\},$$

where $\mathcal{O}^6$ is the set of the 1000 most frequent object categories in the Visual Genome dataset (Krishna et al., 2017) and $\eta_{\mathrm{obj}} \in \mathbb{R}$ is a margin. If the word, however, is an attribute, then the following loss function is minimized instead,

$$\mathcal{L}_{\mathrm{attr}}\left(f_\theta, g_\phi\right) := \sum_{t \in \mathcal{T}}^{|\mathcal{T}|} \mathbb{1}[t \in \mathcal{T}]\left(1 - \Gamma\left(t\right)\right) \log[\sigma\left(\boldsymbol{x}_i^\top g_\phi\left(\mathcal{T}\right)_t\right)] + \mathbb{1}[t \neq \mathcal{T}]\Gamma\left(t\right) \log[1 - \sigma\left(\boldsymbol{x}_i^\top g_\phi\left(\mathcal{T}\right)_t\right)],$$

where $\sigma : \mathbb{R} \mapsto [0, 1]$ is a sigmoid activation function, $\Gamma\left(t\right)$ is the fraction of positive samples for attribute $t$ in a mini-batch, and $\mathcal{T}^7$ denotes the set of the 1000 most frequent attribute categories in the Visual Genome dataset (Krishna et al., 2017).

### 4.3.3 Increasing representational alignment (Figure 1i)

The distillation of knowledge from a teacher network into a student network is a powerful tool in machine learning. It is used to a) compress a large teacher network into a smaller (and faster) student model, b) transfer knowledge from one modality to another (e.g. RGB to depth images), and c) combine the knowledge from an ensemble of teachers into a single student network. While initial work in this area Hinton et al. (2015) focused on behavioral alignment, Tian et al. (2019) proposed a general framework for transferring knowledge by aligning (intermediate) representations. Their setup for transfer between modalities (b) is as follows:

- **Data** $\mathcal{D}$**:** Let $\mathcal{D} := \{(s_i, r_i)\}_{i=1}^n$ be a dataset of $n$ pairs of different modalities (e.g., RGB and depth images).

- **System** A: Let $f_\theta : \mathbb{R}^{H \times W \times C_1} \mapsto \mathbb{R}^p$ be any (pretrained) neural network function parametrized by $\theta$ that maps a set of inputs (e.g., RGB images) to a set of $p$-dimensional outputs and takes the role of the teacher network.

  – **Its measurements** $X$**:** Let $\boldsymbol{X} \in \mathbb{R}^{n \times p}$ be the matrix of stacked activations extracted from a layer/module of the neural network function $f_\theta$ for all $n$ RGB images where $\boldsymbol{X} := (f_\theta\left(s_1\right), \ldots, f_\theta\left(s_n\right))$.
  – **Its embedding** $V$**:** Let $\boldsymbol{V} = \boldsymbol{X}$.

- **System** B: Any trainable neural network function that takes the role of the student network. Let $g_\phi : \mathbb{R}^{H \times W \times C_2} \mapsto \mathbb{R}^d$ be another deterministic neural network function parametrized by $\phi$ that maps a different set of inputs (e.g., depth images) to a set of $d$-dimensional outputs.

  – **Its measurements** $Y$**:** Let $\boldsymbol{Y} \in \mathbb{R}^{n \times d}$ be the matrix of stacked activations extracted from a layer/module of the neural network function $g_\phi$ for all $n$ depth images where $\boldsymbol{Y} := (g_\phi\left(r_1\right), \ldots, f_\phi\left(r_n\right))$.

---

[6] Since we deal with non-contextual word representations, here, we can simply treat $\mathcal{O}$ as a sequence of words rather than a set and apply the neural network function $g_\phi$ (sequentially) to it.

[7] Again, here we can treat $\mathcal{T}$ as a sequence of words rather than a set and apply $g_\phi$ to the sequence to obtain a representation for each attribute word.

– **Its embedding $W$:** Let $\boldsymbol{W} = \boldsymbol{Y}$.

- **Differentiable and directional alignment function $\delta(V, W)$:**

$$\delta(V, W) = \max_h \mathcal{L}_{\text{critic}}(g_\phi, h)$$
$$= \mathbb{E}_{P(X,Y)} \left[ \log h(\boldsymbol{x}, \boldsymbol{y}) \right] + N \mathbb{E}_{P(X)P(Y)} \left[ \log(1 - h(\boldsymbol{x}, \boldsymbol{y})) \right].$$

Here $h : \mathbb{R}^p \times \mathbb{R}^q \mapsto [0, 1]$ is a differentiable function that is trained alongside the student. Thus in this case, the alignment function $\delta(V, W)$ is not fixed but instead fitted to the teacher and student networks $f_\theta$ and $g_\phi$. Note that the two expectations are taken over sampling matching pairs of inputs (i.e. $(s_i, r_i)$) and over non-matching pairs of inputs (i.e. $(s_i, r_j)$ with $i \neq j$) respectively. The factor $N$ is a hyperparameter that determines the relative frequency of non-matching pairs with respect to matching pairs. Tian et al. (2019) show that in this setup $\mathcal{L}_{\text{critic}}$ is a lower bound on the mutual information $I(\boldsymbol{V}; \boldsymbol{W})$.

## 5 Open problems & challenges in representational alignment

In the previous sections, we have presented a unifying framework for analyzing representational alignment that encompasses a wide range of research disciplines. We highlighted commonalities in the work being pursued by researchers across these fields: despite their seemingly disparate natures, each field is conducting profound inquiries into representational alignment and researchers from each field bring complementary perspectives to the table.

We next look ahead and outline a series of challenging unsolved questions that transcend these disciplines. We hope that by identifying these shared challenges, we promote a holistic approach to problem-solving that can catalyze inter-disciplinary collaboration and lead to further progress: not just in each individual field, but across them (and perhaps even sparking new sub-disciplines). We encourage an exchange of ideas and perspectives among our diverse scientific communities, whose combined efforts are well-positioned to help unravel the complexities of representational alignment and advance the design of more representation-aligned information processing systems.

### 5.1 Selecting data and stimuli

Any attempt to either measure or increase representational alignment begins with selecting the dataset $\mathcal{D}$ over which to compute alignment. The degree of alignment measured, or the results of increasing alignment, can depend dramatically on the dataset used.

In particular, if the dataset over which representation alignment is computed is too restricted, the results may not generalize. For example, various features may be confounded in naturalistic data, which can lead to overestimating alignment between models that rely on different features (e.g. Malcolm et al., 2016; Groen et al., 2018; Dujmović et al., 2022). For example, the strong correlation between shape and texture in natural photos may mask the extent to which humans and CNNs rely on distinct features for object recognition (Landau et al., 1988; Baker et al., 2018; Geirhos et al., 2019; Hermann et al., 2020; though cf. Jagadeesh and Gardner, 2022). Likewise, selecting natural stimuli to test an effect of a single feature can introduce biases in other correlated features (Rust and Movshon, 2005)—for example, confounds between lower-level statistical features like Fourier power and more conceptual features like subjective distance or object category can make it harder to identify which is driving neural activity from natural images (Lescroart et al., 2015).

On the other hand, it may be invalid to draw certain inferences based on representations of overly simplistic, even if carefully controlled, stimuli. For example, processing naturalistic stimuli, such as reading a long, continuous text, may engage fundamentally different processes than more controlled tasks over shorter stimuli (Hasson et al., 2015). As a more concrete example, retinal neurons were originally studied with simple bar and grating stimuli; however, some retinal neurons are sensitive to more complex interactions of features, such as foreground motion against a moving background (Ölveczky et al., 2003). Thus, there are dramatic representational differences on datasets of naturalistic stimuli (Karamanlis et al., 2022). Research in machine learning has similarly shown that studying model representations in the context of one dataset may suggest that neurons encode a particular type of feature that is quite different than what appears to be encoded when

studying representations in the context of a different dataset. For example, neurons in the language model BERT (Devlin et al., 2018) appear to encode song titles given one dataset, but dates of historical events given another (Bolukbasi et al., 2021). Thus, the interpretations we draw from our analyses may be biased by the limitations of the data we consider. This issue is not restricted to sparse coding: similar issues can arise under distribution shifts when using RSA or other distributed representation analyses (Dujmović et al., 2022; Friedman et al., 2024). Thus, it is important to assess representational similarity on as diverse a dataset as possible — ideally one that includes both naturalistic stimuli, and more controlled ones that explicitly reduce confounding among important features (Rust and Movshon, 2005; Bowers et al., 2022; Hermann et al., 2023) — and to test on held-out categories of stimuli, in order to determine the generality of the analysis.

However, as noted above (§2.2.5), representational alignment and dataset selection can be mutually reinforcing. Representational alignment can be used to identify key cases where models disagree, by synthesizing optimally "controversial stimuli' that maximally distinguish between the representation spaces (Golan et al., 2022; Groen et al., 2018), or even by selecting the most controversial among large sets of natural stimuli (Hosseini et al., 2024a), which can then be tested on humans or animals. Likewise, representational alignment can be used to optimize stimuli that drive a particular response (Tuckute et al., 2023). The mechanisms of alignment can then be diagnosed through controlled experiments that manipulate stimulus factors (e.g. Opielka et al., 2024). Thus, there can be a virtuous cycle in which measuring representational alignment allows for better selection of datasets that support precisely measuring and understanding representational alignment, and so on. These investigations demand a multidisciplinary perspective drawing on data collection and experimentation practices across research communities.

## 5.2  Defining, probing, and characterizing representations

Once we have chosen systems to compare, and stimuli over which to compare them, we must decide how to present the stimuli to them and how to extract representations. For example, human image processing is recurrent and in some cases this computation can produce more accurate representations over time; thus in some cases non-recurrent network behavior may appear similar to humans under time pressure, but not humans given long times to process a stimulus (e.g. Elsayed et al., 2018)—and presumably some of the underlying representations would reflect this evolution. Thus, details like time of stimulus presentation may in some cases substantially affect the measured representational patterns and similarity between two systems. Likewise, neural representations are dynamic and context-sensitive, and thus presentation order can affect the representation of stimuli. Thus, the presentations format should ideally be designed to align between the two systems as closely as possible, and randomize factors that cannot be aligned.

Extracting representations also poses challenges. For example, in a deep transformer language model, which layers or components (e.g. attention heads or MLPs) should we analyze? If we are interested in human brain activity, how should we record it? Indirect measures like fMRI or EEG can distort or enhance features compared to the information that is computationally available to the underlying system (Ritchie et al., 2019). Or, if we record single-cell neural activity from cortical cells, which regions should we target? These decisions can radically change the results of the analysis. For example, certain kinds of knowledge may be localized in particular regions or components in natural (Kanwisher et al., 1997) and artificial (Manning et al., 2020; Meng et al., 2022) neural networks. Which regions should we study?

Ideally, we would compute representations over all regions and components of each system, and compare these pairwise. Pairwise comparison can reveal similarities in processing, such as parallels in progression through regions of the visual cortex and artificial CNNs (Yamins and DiCarlo, 2016). However, it is often experimentally or computationally infeasible to do these analyses in full. Often, it is necessary to rely on the prior literature—and the available tools—to constrain the hypothesis space of representations to consider. Conversations amongst researchers spanning varied disciplines can ensure such choices are well-informed. However, even once we have selected a method of extracting representations, understanding the role that these representations play in computation remains conceptually challenging, as we discuss in section 5.2.2.

### 5.2.1  Eliciting representations from black-box systems

How do we measure the representational alignment of black-box systems whose inner workings we cannot access? One technique that we described above is collecting similarity judgments, but there are often cases

where running similarity experiments is not feasible, e.g., when we work with a high dimensional and large dataset (however, see Marjieh et al. (2023a) for some recent progress in this direction). An alternative is based on Markov Chain Monte Carlo (MCMC) sampling processes that are widely used in machine learning and physics (Metropolis et al., 1953; Hastings, 1970). The method was first introduced by Sanborn and Griffiths (2007) where participants gradually refined high-dimensional objects by acting as the rejection function in an MCMC sampling chain. Under specific conditions that can be empirically validated, this method converges to a sample from the hidden distribution or representational prior of the participants (Sanborn et al., 2010).

Another similarly adaptive technique is serial reproduction (Xu and Griffiths, 2010; Langlois et al., 2017). This method employs a Gibbs sampling algorithm where participants are tasked with directly recalling and replicating intricate objects, effectively sampling from the underlying prior. Examples include the reproduction of rhythmic sequences (Jacoby and McDermott, 2017; Jacoby et al., 2021a), melodies (Anglada-Tort et al., 2023), or specific spatial positions shown to the participants (Langlois et al., 2021b). This methodology is especially potent in areas where the black-box system, in this instance, a human, can reproduce intricate objects without intermediaries. A recent advancement by Harrison et al. (2020) suggests a technique for modifying object dimensions by interacting with it using a computer slider. Using the Gibbs sampler, this approach has been instrumental in deriving foundational semantic "prototypes" for facial structures (Harrison et al., 2020), emotional prosody (Van Rijn et al., 2021; van Rijn et al., 2022), visual patterns (Kumar et al., 2022), and musical chords (Marjieh et al., 2024a).

It is worth noting that while these methods predominantly involve human subjects, there is a significant overlap with machine learning generative paradigms. Indeed, Marjieh et al. (2023b) have recently demonstrated the mathematical parallels between serial reproduction and diffusion processes. This connection hints at the promising potential of representation elicitation methods in enhancing the interpretability of machine learning, as well as fostering generative models that better resonate with human preferences in forthcoming research.

### 5.2.2   The relationship between representation and computation

In general, we are interested in understanding (or modifying) the representational structure of a system in order to understand (or modify) more abstract computations. However, this raises a thorn for representational alignment research: our methods and interpretation of results depend upon the complex relationship between representation and computation (cf. Churchland and Sejnowski, 1988). Here, we highlight some challenges and questions about this relationship.

**Extraneous influences on representations:** Representations may be shaped by other implementation-level factors that are not essential to the computational process. For example, biological representations may be constrained by energetic demands (e.g., Laughlin, 2001), while deep learning representations may be biased by which features are already represented before training, or which are learned more readily (Hermann and Lampinen, 2020; Farrell et al., 2023; Lampinen et al., 2024). These extraneous factors may cause us to either under- or overestimate representational similarity between systems with different learning processes and implementations (Dujmović et al., 2022; Griffiths et al., 2023; Friedman et al., 2023).

**Context-dependent & dynamic representation:** Biological neural representations are dynamic and contextual; they change with repetition (Grill-Spector et al., 2006), attention (Cukur et al., 2013; Birman and Gardner, 2019), context (Brette, 2019; Deniz et al., 2023), and time (Rule et al., 2019). When performing representational similarity analysis, we are forced to treat a single representation (or a within-participant average) as though it were a canonical representation of that stimulus. However, this inevitably elides important details of the dynamic role each representation plays in the system's computation.

**Philosophical issues in representation and computation:** The practical issues above hint at deeper philosophical issues. Representational alignment is grounded in a computational perspective on natural intelligence, particularly, the notion that a system must necessarily form representations of its inputs in order to produce intelligent behavior. This perspective underlies, for example, the idea that there exists an embedding "function" that can be mapped across a set of stimuli to produce a tensor of embeddings.

However, other perspectives de-emphasize representation and computation in favor of the dynamic interaction between an intelligent system and its environment (e.g., Brooks, 1991; Cisek, 1999). From such perspectives, measuring alignment between tensors of "representations" may seem misguided. Indeed, as noted above, the brain is a dynamical system whose responses to stimuli change and adapt. Thus, how can we philosophically justify aligning "representations" between artificial and natural intelligence?

While we acknowledge the challenges posed by these issues, we take a more *pragmatic* perspective on representation (cf., Poldrack, 2021; Cao, 2022; Cao and Yamins, 2024) and interpret a system's internal responses as representations insofar as they play a "representation-like" role in its behavior. The empirical evidence that aligning representations of neural networks to human ones can improve generalization and transferability (e.g., Muttenthaler et al., 2023b) helps to justify this approach. However, we believe that more deeply analyzing the dynamic role of the system's internal responses in its behavioral interactions could yield greater insights, or greater ability to align systems. Indeed, some recent works are moving in this direction; for example, Ostrow et al. (2024) propose a Dynamical Similarity Analysis (DSA) method that focuses on temporal dynamics, and find that it more accurately identifies similarities among recurrent networks on various tasks. Additional investigations confirm that DSA, as a metric developed with dynamical representations in mind, is better at identifying computationally relevant representations in RNNs than metrics which were conceived for static representations but can be adapted to capture dynamics Guilhot et al. (2024).

### 5.3 Measuring alignment

There are also challenges in measuring alignment between systems. As noted above (§3.3.4), different measures of similarity have distinct advantages and disadvantages. For example, we may be interested in asymmetries that are obscured by symmetrical metrics, or we may want to evaluate how fitting parameters in the alignment changes conclusions. In many cases, different metrics can yield different conclusions about the relationship between two systems (e.g. Minnema and Herbelot, 2019; Cloos et al., 2024a). Thus, as noted above, it is useful to compare systems using multiple metrics.

Yet, there are also shared challenges across similarity measures that are more difficult to address, again due to the complex relationship between representation and computation. For example, similarity metrics generally impose the assumption that smaller differences between two representations are less important than larger ones. For example, (unregularized) linear regression, or RDMs computed with Euclidean distance, assume that the squared distance between two representations measures how important the distinctions between them are. However, this may not always be a good assumption. Sometimes even if a system represents two signals equally well, and uses them equally often, one will carry much less variance — i.e., changes in the signal will result in smaller changes in the representations as measured by Euclidean distance metrics — perhaps due to inductive biases or learning dynamics (Lampinen et al., 2024). Unless we have some way of knowing how "important" different aspects of a representation are to each system's computations, and accordingly adapting our similarity measures, our measures of representational alignment will fail to perfectly capture the underlying computational similarity.

### 5.4 Will representational alignment help improve the alignment of behavior?

Representational alignment focuses on *the representation space of a system*; i.e., the activations yielded by the information processing function of a system (see §3). However, as noted above (§5.2.2), the relationship between representation and computation is complex. The outputs of systems can be aligned even if these systems have different representations, and vice versa (e.g. Hermann and Lampinen, 2020; Davari et al., 2023; Conwell et al., 2023; Cloos et al., 2024a; Bo et al., 2025); likewise, systems that have similar representations early in processing may diverge in later regions to produce different outputs (Singer et al., 2022). Thus, representational alignment between systems is not a prerequisite for aligned outputs, nor will it guarantee them.

However, initial representations constrain what a system will learn to output, and conversely, what a system learns to output will shape its representations. Thus, although it may be possible to achieve output alignment without representational alignment, the tight coupling between representations and outputs motivates studying representational alignment as one potential tool for achieving output alignment. Representational alignment

could help researchers to pinpoint potential causes for output (mis-)alignment of systems, and could be used as a complement to more direct strategies for improving output alignment (Peterson et al., 2018; Barrett et al., 2018; Toneva and Wehbe, 2019; Fel et al., 2022; Muttenthaler et al., 2023a;b; Fu et al., 2023), which may be especially important when designing human-centric AI thought partners (Collins et al., 2024b).

### 5.5 Possible risks of representational alignment

It is worth noting that there may be risks to optimizing AI systems for representational alignment. For instance, increased representational alignment could potentially make it more difficult to detect that digital artifacts or communications (e.g., text, video, conversation, etc.) are produced by AI systems rather than humans. In the case of aligning with a biological system, it is paramount to consider which systems (e.g., which humans) we do or do not wish to align towards, and what downstream biases could occur as a result of these potentially implicit design choices (cf. Gabriel, 2020). We encourage further work to characterize possible risks and develop frameworks to guard against such possible negative ramifications.

## 6 Conclusion

Representational alignment is increasingly central to the various fields that study information processing, including cognitive science, neuroscience, and machine learning. In each field, researchers attempt to *measure* the alignment between representations from different systems, to *bridge* between distinct systems by bringing their representations into a shared space, and to *increase* the representational alignment of two systems. However, there is no clear common language for discussion between these different communities; thus, researchers are often unaware of related ideas, methods, and empirical results. In this Perspective, we have attempted to build bridges to help align terminology and methods across these fields, and to highlight some of the history and recent developments within each. We hope that our work will simultaneously increase the sharing of related ideas and methods across fields, and raise awareness of common challenges and open questions. More broadly, we hope that seeing the varied perspectives outlined here will inspire other researchers to apply the ideas and tools of representational alignment to understanding or building (more) intelligent systems.

## Acknowledgments

LM and KRM acknowledge funding from the German Federal Ministry of Education and Research (BMBF) for the grants BIFOLD22B and BIFOLD23B. KRM was supported in part by the Institute of Information & Communications Technology Planning & Evaluation (IITP) grant funded by the Korea government (MSIT) (No. RS-2019-II190079, Artificial Intelligence Graduate School Program, Korea University) and grant funded by the Korea government (MSIT) (No. RS-2024-00457882, AI Research Hub Project). KMC acknowledges support from the Marshall Commission and Cambridge Trust. AW acknowledges support from a Turing AI Fellowship under grant EP/V025279/1, The Alan Turing Institute, and the Leverhulme Trust via CFI. This work was supported by an NSERC fellowship (567554-2022) to IS. JA acknowledges funding through a Medical Research Council intramural programme (MC_UU_00030/7), a Gates Cambridge Scholarship through the Bill and Melinda Gates Foundation, and additional support through Intel Labs. We thank Mike Mozer, Alex Williams, Rose Cao, Todd Gureckis, and many members of Gureckis Lab for their excellent comments on an earlier version of this manuscript.

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
