# OpenReview forum: "Getting aligned on representational alignment"
_TMLR — Accepted by TMLR_

### Review · Reviewer_CMuC · 2025-06-29

**Summary Of Contributions:**

The paper proposes a theoretical framework that could be used to formulate and approach representation alignment problems coming from different fields.

The paper systematizes common classes of problems that re-emerge in representation alignment research across different fields: measuring the alignment of different systems, bridging the representations of different systems, and increasing their representation alignment, and reviews them in the context of machine learning and cognitive- and neuro-sciences.

The proposed framework can be used to formulate there problems in a unified way across fields; the authors scrutinize different similarity metrics that can be used to measure the alignment in different settings, and show examples of how previously published research can be translated to the language of the proposed framework.

**Audience:**

Yes

**Broader Impact Concerns:**

None.

**Claims And Evidence:**

Yes

**Requested Changes:**

I have only a few non-critical stylistic issues.

3.3.1 Similarity or dissimilarity quantifying: Review this sentence: “Similarity quantifying functions that have been transformed into distances, such as the cosine distance—or, equivalently, 1 - the Pearson correlation coefficient—are better suited to measure representational alignment.” ‘1 - the Pearson correlation coefficient’ is a little confusing, I did not render it as “one minus the Pearson correlation coefficient” at first.


3.1 High-level overview (b): In the sentence “…or example the neuronal activity through the entire brain during preconception…” ‘preconception’ seems out of place.

**Strengths And Weaknesses:**

Strengths:

Profound multidisciplinary review of representation alignment works and examples of how to reformulate them within the proposed framework.
Detailed overview of different similarity measures and their use cases.

Weaknesses:

None.

---

> ### Author Response · Authors · 2025-08-15
>
> Thank you for the positive review and for catching those stylistic issues! We have now updated both of those sentences based on your feedback.

---

### Review · Reviewer_L8p3 · 2025-06-30

**Summary Of Contributions:**

This manuscript presents a comprehensive review of research and a conceptual framework that unifies representational alignment. The authors address key questions regarding how to measure the similarity between representations formed by different systems (e.g., humans, animals, and machines), as well as how representations can be modified to increase alignment.

**Audience:**

Yes

**Broader Impact Concerns:**

No concerns.

**Claims And Evidence:**

Yes

**Requested Changes:**

I think the authors should consider incorporating actionable guidelines that would enable researchers to implement representational alignment techniques more directly. That could also be connected directly to the open problems. It would be great to have some schematics. I feel that there is a significant jump from Chapter 4 to Chapter 5.

**Strengths And Weaknesses:**

The manuscript effectively summarizes a broad range of literature and integrates insights from studies on neuroscience and artificial intelligence. It directly addresses critical methodological questions, clearly discussing representational similarity techniques, which will be particularly useful for researchers aiming to quantify alignment. Additionally, it provides a cohesive conceptual framework and offers a structure to previously fragmented research areas.

The manuscript does not offer an alternative predictive perspective that considers contrastive predictive coding (CPC) [1] or joint embedding predictive architectures (JEPA) [2] as alternatives to alignment.

1) Oord, Aaron van den, Yazhe Li, and Oriol Vinyals. "Representation learning with contrastive predictive coding." arXiv preprint arXiv:1807.03748 (2018).

2) Assran, Mahmoud, et al. "Self-supervised learning from images with a joint-embedding predictive architecture." Proceedings of the IEEE/CVF Conference on Computer Vision and Pattern Recognition. 2023.

---

> ### Author Response · Authors · 2025-08-15
>
> Thank you for the positive review and helpful suggestions! We have added a sentence about the two suggested references to Section 2.3.1., incorporated some more actionable guidelines (our guidelines and suggestions to readers/researchers can primarily be found in Sections 3 and 5), and better emphasized how researchers should interpret and use our framework as summarized by the schematic in Fig 2.

---

### Review · Reviewer_d8Kx · 2025-08-04

**Summary Of Contributions:**

This paper provides a review of representational alignment techniques used in Cog Sci, Neuroscience, and Machine Learning. Several prominent studies are described, along with several representational alignment metrics. For the three domains, the authors lay out methods to measure, bridge, and increase the alignment. Several practical considerations are addressed.

**Audience:**

Yes

**Broader Impact Concerns:**

I do not believe a broader impact section is required for this paper. In any case, the authors lay out possible risks in section 5.5.

**Claims And Evidence:**

Yes

**Requested Changes:**

**Critical Recommendations:**

See 'Other issues' in strengths and weaknesses: Points 2, 3, 5, 8, 10, 11, 12. In general, please improve the presentation to the extent the authors deem reasonable. See also Point 3 (add more details about LLM-Brain alignment) from the list of trends missing in the paper.

All other issues would strengthen the paper, but are not critical.

**Strengths And Weaknesses:**

Strengths:

1) The paper aims to bridge representational alignment studies across three disciplines (CogSci, Neuroscience, Machine Learning).
2) The paper provides an extensive list of papers (> 400) in the field.
3) The paper goes into depth with specific case studies of measuring, bridging, and increasing alignment across the three disciplines.
4) Overall, this paper is a great resource and does a good job of being almost a one-stop shop for learning about representational alignment. I will definitely be using it if I need to look up something.

Weaknesses:

I believe the current presentation makes it useful mainly for people already in the field rather than readers in other fields looking to get into the general alignment space.

More concretely, below is a list of issues that I found in the paper (despite the longer weakness list than the strengths, I believe this is an important contribution to the field and recommend acceptance):

Here are some recent trends in the field that I don’t see get much attention in the paper (granted, several of these are very recent papers/preprints and I don’t hold it against the authors for not being aware of them and **I will not use their inclusion as a criterion to judge their revised manuscript, except point 3**):

1. Recent work on the relation between representational and behavioral similarities (e.g. Cao and Yamins, Prince et al., CCN 2024, Bo et al., CCN 2025) - the authors touch upon this in section 5.4, but these papers could be relevant there.
2. Meta-analyses or theory papers for metric comparison or selection (e.g. several recent papers by Alex Williams’ group, e.g. the one comparing RSA, CKA and CCA; Thobani et al., CCN 2025; Soni et al.)
3. Recent work on LLM-Brain alignment (Fedorenko lab, Huth lab) should be discussed in more detail, which has been getting a lot of attention at venues such as CCN, UniReps workshop at NeurIPS, or the ReAlign workshop that the authors already include.
4. Other adjacent work (which I am not sure where to place but I believe would add to the paper):
’Everything, Everywhere, All at Once: Is Mechanistic Interpretability Identifiable?’ ****and ‘Relating Representational Geometry to Cortical Geometry in the Visual Cortex’.
5. Work on the dynamics of alignment throughout training  - Scholte et al., CCN 2024 talk about early alignment; see also Rossem and Saxe, 2024.
6. Anything about alignment in or with other species?

Other issues:

1. Table 2 is hard to parse in the current form. Most papers are not described anywhere, so it basically is just giving me some statistical sense of what techniques are being used, but not what people are finding using them.
2. In some places, the paper merely points to other papers in the area without explaining why one should be interested in them (in these cases, the paper is not serving a purpose beyond the github repositories listing important papers in an area (no insult intended, I find these repositories very useful)). What insights can a reader draw from **“**In particular, the fact that word representation spaces of words learned by predicting co-occurrence (Mikolov et al., 2013a; Pennington et al., 2014) allowed analogical reasoning by simple linear algebra operations (e.g., king − man + woman = queen),
attracted a great deal of interest and investigations (Ethayarajh et al., 2018).” What did Ethayarajh et al do exactly? I understand space constraints, but the paper becomes hard to parse if the reader has to switch to other papers to know what has been done in an area.
3. Figure 1h is labeled Khosla 2023, while the figure caption calls it Khosla 2022. The figure itself seems to be from Khosla 2022.
4. Section 5.5, first couple of sentences: a little uninformative and a bit of a stretch.
5. The main figure (Fig 1) should be more readable at the default resolution, and instead of just the authors, it would be good to name the technique as well (if applicable).
6. 2.1.2 has some results but would be good to interpret that body of results. if object recognition models that do better show worse performance, what does it say about features driving alignment vs performance?
7. 2.2.6: in the sections on communication, some recent work by the Hasson lab would also be relevant.
8. The section on HSIC (3.3.5) is important. Too often I see people using linear CKA and calling it just CKA. But that section would be improved if there are more details about the pros and cons of the various versions of CKA (or perhaps just when do these implementations disagree).
9. I see shape metrics or even regression based metrics not getting enough attention, especially not commensurate with the latter’s prominence in the field. In terms of laying out other metrics, the review paper by Klabunde does a much better job (which, to the authors’ credit, is cited as a more in-depth study of metrics)
10. 2.3.4 - there’s a huge collection of papers with barely any description and then there’s Rane et al. described in depth separately. Would have been better in my opinion to cite fewer papers but describe what a reader should get out of them.
11. 2.3.5: What is constitutional AI?
12. 2.3.3: “For example, finding circuits (Olah et al., 2020; Nanda et al.,
2023) that qualitatively align with semantic meaning (e.g., curves) could provide insights.” A short description of how they find the circuits would be good.
13. There are 427 papers cited but the reader only gets to learn in about a small percentage of these, most papers are referred to with very little accompanying detail and the reader will need to look everything up themselves, which somewhat defeats its purpose in my opinion.
14. If the authors want to make space for more details, most of the details in chapter four can be condensed into tables because details like Data, System A, System B, Metric are being repeated for all analyses.

---

> ### Author Response · Authors · 2025-08-15
>
> Thank you for the positive review and detailed feedback, we really appreciate the detailed read-through of the manuscript and especially appreciate the suggestions from recent work which we had missed but greatly enjoyed catching up on! We have now implemented these suggestions including a) adding in the suggested additional citations, literature, discussion, and explanations (relating to the points from the weaknesses section, and points 2, 4, 6, 7, 8, 11, 12 from the other issues list), b) fixing figure 1 and updating its caption to be more informative (relating to points 3 and 5 from other issues), c) using the Semantic Scholar API to pull the Semantic Scholar TLDRs (one-sentence summaries) of the majority of the papers we cite (those that could be found programmatically by searching the API) and adding these to the references so that readers have some more information about what each paper actually covers/proposes without having to look it up (relating to points 1, 2, 10, and 13 from other issues). We think the paper is in much stronger shape after this revision and hope that it will indeed be an almost “one-stop shop for learning about representational alignment” as the reviewer so kindly suggested!

---

### Decision · Action_Editor_Wnk5 · 2025-09-21

**Recommendation:** Accept as is

**Audience:**

Yes

**Audience Explanation:**

The paper considers a topic of wide interest to the representation learning community.

**Claims And Evidence:**

Yes

**Claims Explanation:**

This survey paper has been positively evaluated by three independent reviewers and the authors have further enhanced it based on the provided feedback. The already wide scope was expanded to be even more comprehensive and whatever minor inconsistencies were found were corrected. All reviewers agreed that the paper is clearly written and is valuable to the community.